# THE CHOICE OF DIVERGENCE: A NEGLECTED KEY TO MITIGATING DIVERSITY COLLAPSE IN REINFORCEMENT LEARNING WITH VERIFIABLE REWARD

**Long Li**[1,2,†]**, Zhijian Zhou**[1,5,†]**, Jiaran Hao**[2]**, Jason Klein Liu**[2]**, Yanting Miao**[4]**, Wei Pang**[4]**,
**Xiaoyu Tan**[2]**, Wei Chu**[2]**, Zhe Wang**[3]**, Shirui Pan**[3]**, Chao Qu**[1,6,∗]**, Yuan Qi**[1,6]

[1]**Fudan University**      [2]**INFLY TECH**      [3]**Griffith University**
[4]**University of Waterloo**      [5]**Shanghai Innovation Institute**
[6]**Shanghai Academy of Artificial Intelligence for Science**

`seamoke111@gmail.com`

## ABSTRACT

A central paradox in fine-tuning Large Language Models (LLMs) with Reinforcement Learning with Verifiable Reward (RLVR) is the frequent degradation of multi-attempt performance (Pass@k) despite improvements in single-attempt accuracy (Pass@1). This is often accompanied by catastrophic forgetting, where models lose previously acquired skills. Despite numerous proposed methods, the community's focus on the standard reverse-KL divergence has led to a surprising oversight: the potential of alternative f-divergences as a proactive solution has been largely unexamined. We argue that standard RLVR objectives—both those using the mode-seeking reverse-KL divergence and those forgoing a divergence term entirely—lack a crucial mechanism for knowledge retention. The reverse-KL actively accelerates this decay by narrowing the policy, while its absence provides no safeguard against the model drifting from its diverse knowledge base. We propose a fundamental shift in perspective: using the divergence term itself as the solution. Our framework, Diversity-Preserving Hybrid RL (DPH-RL), leverages mass-covering f-divergences (like forward-KL and JS-divergence) to function as a 'rehearsal mechanism'. By continuously referencing the initial policy, this approach forces the model to maintain broad solution coverage. Math and SQL generation experiments show that DPH-RL surpasses the GRPO baseline by improving both in-domain Pass@1 and Pass@k scores, and effectively prevents catastrophic forgetting on out-of-domain tasks. Additionally, DPH-RL is more training-efficient because it computes f-divergence using generator functions, requiring only sampling from the initial policy and no online reference model. Our work highlights a crucial, overlooked axis for improving RLVR, demonstrating that the proper selection of a divergence measure is a powerful tool for building more general and diverse reasoning models. Code and data are available at: `https://github.com/seamoke/DPH-RL`.

## 1 INTRODUCTION

Reinforcement Learning with Verifiable Rewards (RLVR) has recently shown significant success in enhancing the mathematical and coding capabilities of Large Language Models (LLMs) (OpenAI, 2023; Team, 2024; Yang et al., 2025; Meta AI, 2024; Guo et al., 2025). Despite this progress, a critical paradox has emerged: while RLVR-tuned models consistently improve the probability of generating a correct solution in a single attempt (Pass@1), their performance when multiple attempts are permitted (Pass@k) often stagnates or even degrades compared to their base models (Yue et al., 2025). This discrepancy suggests that RLVR, rather than teaching novel reasoning, may instead overfit the model to known solution paths, thereby sacrificing diversity.

To counteract the narrowing of the model's output distribution (Wang et al., 2025; Yue et al., 2025), which reduces solution diversity, researchers have explored several parallel strategies. One major

---

∗Corresponding author. † Equal contribution.

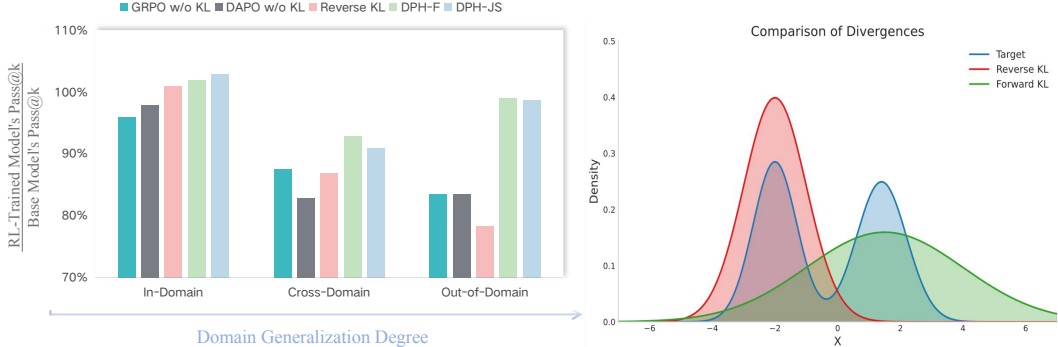

Figure 1: The left panel evaluates the performance gap in Pass@k between the RL-trained model and the Base Model across test sets with varying degrees of divergence from the training data. The right panel visualizes the distributions of reverse-KL and forward-KL.

line of work attributes the phenomenon to entropy collapse and encourages exploration by controlling entropy (Cui et al., 2025; Cheng et al., 2025; Liang et al., 2025). Another distinct approach focuses on directly optimizing the Pass@k metric (Mahdavi et al.; Walder & Karkhanis, 2025), which serves as a more direct proxy for solution diversity than entropy. A third strategy focuses on training setups, such as using extra data (Yan et al.; Dong et al., 2025a), fine-tuning hyperparameters (He et al., 2025; Yu et al., 2025), or employing a hybrid RL+supervised fine-tuned (SFT) training paradigm (Liu et al., 2025b). Alongside these methods, a fourth, more fundamental component of the RL objective—the Kullback-Leibler (KL) divergence term used to constrain policy updates—also plays a crucial role.

However, while the first three avenues have been extensively explored, the influence of the KL divergence term has remained largely under-examined. The community has almost universally adopted the standard **reverse-KL divergence** ($D_{\mathrm{KL}}(\pi_\theta||\pi_{ref}) = \mathbb{E}_{\pi_\theta} \log \frac{\pi_\theta}{\pi_{ref}}$) (Kazemnejad et al., 2024; Schulman et al., 2017; Shao et al., 2024), whose well-established **mode-seeking** nature (Bishop & Nasrabadi, 2006) theoretically forces the policy to converge on a single high-probability solution. Our experiments confirm this: a reverse-KL objective yields a single solution style, while a forward-KL objective generates multiple styles (Figure 4). Additionally, we observe that models trained with a reverse-KL term or *no KL* term at all suffer from a decline in Pass@k and catastrophic forgetting. As shown in the left panel of Figure 1, both GRPO without KL (Shao et al., 2024), DAPO (Yu et al., 2025), and Reverse KL show a significant decline in Pass@k performance. In out-of-domain scenarios, the Reverse KL approach performs even worse. These models only correctly answer about 85% of the problems they were previously able to solve (Figure 3). This phenomenon is a known challenge in sequential learning paradigms with neural networks, not a fundamental flaw of RL, which in principle does not suffer from this in tabular cases (Hamadanian et al., 2025).

Despite these significant issues, prior RLVR methods have almost exclusively relied on the standard **reverse-KL divergence**, while the exploration of other f-divergences has been largely confined to offline RL or alignment tasks ("f-PO" and "f-DPO") (Wang et al.; Han et al., 2025). This trend severely underestimates their potential to solve the diversity collapse problem. To bridge this critical gap, we introduce the Diversity-Preserving Framework (DPH-RL), an approach that employs different f-divergences to preserve model diversity within the online RLVR framework. A straightforward example is the **forward-KL divergence** ($D_{\mathrm{KL}}(\pi_{ref}||\pi_\theta) = \mathbb{E}_{\pi_{ref}} \log \frac{\pi_{ref}}{\pi_\theta}$), whose theoretical **mass-covering** property penalizes the policy for failing to cover all solutions in the reference distribution. From a practitioner's standpoint, the forward-KL objective effectively creates an "anchor dataset", forcing the model to continuously rehearse its original knowledge base—a mechanism that mirrors human learning and prevents the catastrophic forgetting seen with reverse-KL. We generalize this concept to the broader family of **f-divergences**, including Jensen-Shannon (JS) divergence and $\alpha$-divergence (see Table 5). Our experiments demonstrate that these mass-covering divergences, specifically forward-KL and JS-divergence, lead to significant improvements in both Pass@1 and Pass@k, maintain strong out-of-domain performance, and achieve these gains without external models. Notably, this approach is **orthogonal** to existing methods that focus on entropy control or reward shaping.

We make three main contributions:

1. Systematic Analysis of Diversity Collapse: Focusing on the KL divergence term, we provide a systematic analysis of the solution diversity collapse in RLVR, identifying the standard reverse-KL divergence as a primary cause. We show that its mode-seeking nature not only suppresses Pass@k performance—often to levels below the base model—but also exacerbates catastrophic forgetting and leads to poor out-of-domain generalization.

2. A Novel DPH-RL Framework: We reframe the role of the KL divergence, proposing its use not as a mere policy constraint but as an active diversity-preserving mechanism. Based on this principle, we introduce DPH-RL, a novel framework that employs mass-covering f-divergences (e.g., Forward-KL and JS-divergence) to serve as a rehearsal mechanism. It effectively outperforms GRPO in both Pass@1 and Pass@k, while mitigating the Pass@k degradation relative to the base model.

3. Extensive Empirical Validation: Through extensive experiments on a range of models (Llama and Qwen, 7B to 32B) and complex reasoning tasks (mathematics and SQL), we demonstrate the robustness and superiority of DPH-RL. Our method consistently outperforms prior work on both in-domain and out-of-domain benchmarks, successfully mitigating the trade-off between greedy performance and solution diversity.

## 2 RELATED WORK

A more recent development is RLVR (Yue et al., 2025), a promising strategy for boosting LLM reasoning, especially in areas such as mathematics, coding, and analysis (Shao et al., 2024; Guo et al., 2025; Team, 2024; Lambert et al., 2025; Li et al., 2024c; Shen & Zhang, 2025; OpenAI, 2024; Team et al., 2025; Yang et al., 2025). To address diversity collapse in RLVR, one major line of research focuses on either directly controlling model entropy (Cui et al., 2025; Prabhudesai et al., 2025; Agarwal et al., 2025; Li et al., 2025; Cheng et al., 2025; Liang et al., 2025) or strategically capturing high-entropy tokens (Wang et al., 2025; Chen et al., 2025; MiniMax et al., 2025). However, some research suggests that the entropy of specific tokens is more strongly correlated with model diversity than sentence-level entropy (Wang et al., 2025; Cheng et al., 2025; Deng et al., 2025; Chen et al., 2025); consequently, other methods maintain training diversity by controlling the model's Pass@k metric (Dong et al., 2025b; Zhang et al., 2025b). From a training design perspective, methods such as dynamically adjusting rollout-related hyperparameters (An et al., 2025; He et al., 2025; Yu et al., 2025) or using a hybrid SFT+RL (Liu et al., 2025b;a;c) training paradigm are also employed to prevent diversity collapse. It is worth noting that theoretically, DPH-F can be viewed as a divergence-based Experience Replay, sharing a similar philosophy with some existing work Ouyang et al. (2022); Zhang et al. (2025a), while DPH-JS extends this into a *Regularized Rehearsal* mechanism via dual-sided sampling. Please refer to Appendix D for details.

Our work is most closely related to recent studies that explore f-divergences for policy optimization. Notably, Wang et al. (2024); Han et al. (2025) **replace the reverse-KL divergence in the offline DPO objective with a generalized f-divergence, deriving a new family of "f-PO" algorithms**. However, our work is distinct: unlike f-PO's focus on offline preference alignment, we utilize online RLVR with verifiable rewards. Our objective is to address Pass@k diversity collapse through mass-covering f-divergences rather than human preference alignment.

## 3 PRELIMINARIES

### 3.1 $f$-DIVERGENCE

In information theory, an *f-divergence* is a function $D_f(p\|q)$ that measures the difference between two probability distributions $p$ and $q$. Given a convex function $f : \mathbb{R}^+ \to \mathbb{R}$ such that $f(1) = 0$, the f-divergence is defined as:

$$D_f(p\|q) = \int q(x) f\left(\frac{p(x)}{q(x)}\right) dx$$

. This general form allows for the unification of many common divergence measures under a single theoretical framework. The condition $f(1) = 0$ ensures that $D_f(p\|q) = 0$ if and only if $p = q$. Many common divergences are special cases of f-divergences, corresponding to a specific choice for the generator function $f$. For instance, KL divergence and JS divergence are all instances of f-divergences. The generator function for JS divergence is $(u \log u - (u+1) \log \frac{u+1}{2})/2$. A summary of some prominent f-divergences and their corresponding generator functions is provided in 5.

## 3.2 MARKOV DECISION PROCESS

We consider a discounted Markov Decision Process (MDP) defined by the tuple $(\mathcal{S}, \mathcal{A}, P, r, \rho_0, \gamma)$, where $\mathcal{S}$ is the state space, $\mathcal{A}$ is the action space, $P(s'|s, a)$ is the transition probability, $r(s)$ is the reward function, $\rho_0$ is the initial state distribution, and $\gamma \in [0, 1)$ is the discount factor. A stochastic policy $\pi(a|s)$ defines a probability distribution over actions for each state. The goal is to maximize the performance objective, i.e., the expected cumulative discounted reward:

$$J(\pi) = \mathbb{E}_{\tau \sim \pi} \left[ \sum_{t=0}^{\infty} \gamma^t r(s_t) \right], \quad \text{where} \quad s_0 \sim \rho_0(s_0),\ a_t \sim \pi(a_t|s_t),\ s_{t+1} \sim P(s_{t+1}|s_t, a_t). \tag{1}$$

A seminal result from Kakade & Langford (2002) expresses the performance of a policy $\pi$ in terms of an older policy $\pi_{\text{old}}$ using the advantage function $A_{\pi_{\text{old}}}(s, a)$: $J(\pi) = J(\pi_{\text{old}}) + \mathbb{E}_{\tau \sim \pi} \left[ \sum_{t=0}^{\infty} \gamma^t A_{\pi_{\text{old}}}(s_t, a_t) \right]$. This can be rewritten using the discounted state visitation distribution, $\rho_\pi(s) \triangleq \sum_{t=0}^{\infty} \gamma^t P(s_t = s|\pi)$, which gives the probability distribution over states encountered under policy $\pi$:

$$J(\pi) = J(\pi_{\text{old}}) + \sum_{s \in \mathcal{S}} \rho_\pi(s) \sum_{a \in \mathcal{A}} \pi(a|s) A_{\pi_{\text{old}}}(s, a). \tag{2}$$

Directly optimizing this expression is difficult because the state distribution $\rho_\pi$ depends on the new policy $\pi$. Therefore, algorithms like TRPO and PPO optimize a surrogate objective $L_{\pi_{\text{old}}}(\pi)$ by approximating $\rho_\pi$ with the distribution from the old policy, $\rho_{\pi_{\text{old}}}$:

$$L_{\pi_{\text{old}}}(\pi) = J(\pi_{\text{old}}) + \sum_{s \in \mathcal{S}} \rho_{\pi_{\text{old}}}(s) \sum_{a \in \mathcal{A}} \pi(a|s) A_{\pi_{\text{old}}}(s, a). \tag{3}$$

This approximation is reliable when $\pi$ is close to $\pi_{\text{old}}$, forming the basis of modern policy gradient methods.

## 4 METHOD

In this section, we detail our implementation of reinforcement learning (RL) with an f-divergence regularizer. Our objective is to find the optimal policy for the following optimization problem:

$$\max_{\pi_\theta} \mathbb{E}_{q \sim \mathcal{D}} \left[ \mathbb{E}_{a \sim \pi_\theta(\cdot|q)}[r(a|q)] - \eta D_f(\pi_\theta(\cdot|q) || \pi_{ref}(\cdot|q)) \right]. \tag{4}$$

In this formulation, $\pi_{ref}$ serves as the initial reference policy, such as a base or SFT model. The term $r(a|q)$ represents the reward for taking action $a$ in response to query $q$, and the hyperparameter $\eta$ controls the penalty for deviating from $\pi_{ref}$. While standard algorithms like PPO can optimize the reward term, our discussion focuses on estimating the f-divergence term. A uniform application of this approach across all queries in dataset $\mathcal{D}$ can be suboptimal. For queries where $\pi_{ref}$ performs well, aggressive reward maximization is unnecessary and risks degrading performance. Conversely, for challenging queries where $\pi_{ref}$ struggles, the f-divergence term can overly constrain the policy $\pi_\theta$, limiting its ability to explore better solutions. To address this, we propose a targeted strategy that partitions the dataset $\mathcal{D}$ into two subsets. The methodology is divided into two phases: a **pre-sampling stage** and an **online training stage**.

### 4.1 PRE-SAMPLING STAGE

We consider a specific instance of f-divergence, **Forward-KL**, defined as:

$$D_{\text{forward-KL}}(\pi_\theta || \pi_{ref}) \triangleq D_{\text{KL}}(\pi_{ref} || \pi_\theta) = \mathbb{E}_{a \sim \pi_{ref}}[\log(\pi_{ref}(a|q)) - \log(\pi_\theta(a|q))].$$

To facilitate computing this expectation, we adopt a pre-sampling strategy. Before training $\pi_\theta$, we partition the dataset $\mathcal{D}$. For each query $Q$, we generate and evaluate $k$ independent samples. Based on a correctness threshold, each query is classified as either "**near-perfect**" or "**exploration**," splitting $\mathcal{D}$ into $\mathcal{D}_{\text{pef}}$ and $\mathcal{D}_{\text{exp}}$. This partition allows our agent to focus on challenging examples in $\mathcal{D}_{\text{exp}}$ while using a KL divergence constraint to maintain performance on mastered examples in $\mathcal{D}_{\text{pef}}$. To mitigate sampling bias, we further refine $\mathcal{D}_{\text{pef}}$. For each query in this subset, we draw one final sample. We only retain the query in $\mathcal{D}_{\text{pef}}$ if this new sample is correct; otherwise, we discard the query or move it to $\mathcal{D}_{\text{exp}}$.

### 4.2 Online Training Stage

In the online training stage, we simultaneously train the model using two distinct loss functions tailored for the $\mathcal{D}_{\exp}$ and $\mathcal{D}_{\text{pef}}$ datasets. For samples from $\mathcal{D}_{\exp}$, we want the model to have maximum freedom for exploration. Conversely, for samples from $\mathcal{D}_{\text{pef}}$, we want the model to retain its capabilities. To achieve this, we employ two key f-divergences: **Forward-KL** and **JS divergence**. Our method is thus divided into two approaches: **DPH-F** for forward-KL and **DPH-JS** for JS divergence. Forward-KL divergence, as we define it, penalizes instances where the reference policy $\pi_{ref}$ assigns a high probability to an action, but the new policy $\pi_\theta$ assigns a near-zero probability. This property encourages the new policy $\pi_\theta$ to maintain coverage of all modes of the reference policy $\pi_{ref}$, thereby preserving its original diversity. JS divergence provides a symmetric and more stable alternative to KL divergence. It encourages the new policy $\pi_\theta$ to maintain high similarity with the reference policy $\pi_{ref}$ while achieving high performance, effectively preventing policy collapse.

#### 4.2.1 Generator-Based Implementation

Our implementation relies on pre-sampling from the reference policy. This approach allows us to compute the divergence term using a static dataset, eliminating the need to run inference with the reference model during the online training loop.

**Loss for $\mathcal{D}_{\exp}$** For challenging samples in $\mathcal{D}_{\exp}$, we remove the KL divergence penalty from the loss function entirely. This allows the model to perform pure policy optimization based solely on the reward signal, enabling more aggressive exploration. Specifically, the loss function for these samples is the standard PPO-clip objective:

$$
\mathcal{L}_{\text{DPH-exp}}(\theta) = -\mathbb{E}_{\substack{q \sim \mathcal{D}_{\exp} \\ o_i \sim \pi_{\theta_{\text{old}}}(\cdot|q)}} \left[ \frac{1}{G} \sum_{i=1}^{G} \frac{1}{|o_i|} \sum_{t=1}^{|o_i|} \min \left( \rho_{i,t}(\theta) \hat{A}_{i,t}, \right. \right.
$$
$$
\left. \left. \text{clip}\left( \rho_{i,t}(\theta), 1 - \varepsilon, 1 + \varepsilon \right) \hat{A}_{i,t} \right) \right] \tag{5}
$$

Please refer to Appendix E for the definitions of the symbols in the formula.

**Loss for $\mathcal{D}_{\text{pef}}$** For all f-divergences, the general formula for calculating the loss on $\mathcal{D}_{\text{pef}}$ is:

$$
\mathcal{L}_{\text{pef}}(\theta) = \mathbb{E}_{q \sim \mathcal{D}_{\text{pef}}} [D_f(\pi_\theta || \pi_{ref})]. \tag{6}
$$

For DPH-F, the loss term is: $\mathbb{E}_{q \sim \mathcal{D}_{\text{pef}}} \left[ \sum_a \pi_{ref}(a|q) \log \left( \frac{\pi_{ref}(a|q)}{\pi_\theta(a|q)} \right) \right]$, where $a$ represents a single response sampled once from the reference policy $\pi_{ref}(\cdot|q)$. For DPH-JS, the loss term is: $\mathbb{E}_{q \sim \mathcal{D}_{\text{pef}}} \left[ \sum_a \pi_{ref}(a|q) \left( \frac{u \log u}{2} - \frac{u+1}{2} \log \left( \frac{u+1}{2} \right) \right) \right]$, where $u = \pi_\theta(a|q)/\pi_{ref}(a|q)$.

#### 4.2.2 Overall Loss Function

The total loss for a given batch is a combination of these two objectives. For a mixed batch, we determine whether the data comes from $\mathcal{D}_{\text{pef}}$ or $\mathcal{D}_{\exp}$ and then calculate the corresponding loss:

$$
\mathcal{L}_{\text{DPH-RL}}(\theta) = \mathcal{L}_{\exp}(\theta) + \eta \mathcal{L}_{\text{pef}}(\theta), \tag{7}
$$

### 4.3 Enhanced Monotonic Improvement Guarantee

In this section, we derive an enhanced monotonic improvement guarantee for TRPO-style algorithms that leverage our proposed method. Our framework is built upon two distinct datasets: $\mathcal{D}_{\text{pef}}$, which stores near-perfect reasoning trajectories, and $\mathcal{D}_{\exp}$, an exploration set for tasks the model has not yet mastered. Central to our analysis is a conditional reference policy, $\pi_{\text{pef}}$, whose definition depends on the data source. For states associated with the **near-perfect dataset ($\mathcal{D}_{\text{pef}}$)**, $\pi_{\text{pef}}$ is the policy induced by the stored correct trajectories. The f-divergence regularization term $D_f(\pi || \pi_{\text{pef}})$ pulls the learned policy $\pi$ towards these expert solutions. For states associated with the **exploration dataset ($\mathcal{D}_{\exp}$)**,

$\pi_{\text{pef}}$ is defined as the current policy $\pi$. This makes the regularization term $D_f(\pi\|\pi)$ equal to zero, thereby disabling it and permitting unrestricted exploration. To analyze the effect of this conditional regularization, our derivation hinges on the following mild assumption.

**Assumption 1.** *For any policy $\pi$ encountered during training, there exists a constant $\delta \geq 0$ such that for any state $s$, the expected advantage of actions from the reference policy $\pi_{pef}$, evaluated with respect to $\pi$, is lower-bounded by $\delta$: $\mathbb{E}_{a_{pef}\sim\pi_{pef}(\cdot|s)}[A^\pi(s, a_{pef})] \geq \delta$.*

This assumption is well-founded. In regions of the state space well-represented by the near-perfect dataset $\mathcal{D}_{\text{pef}}$, the actions from $\pi_{\text{pef}}$ are expected to be superior, yielding a positive advantage ($\delta > 0$). Conversely, for states associated with $\mathcal{D}_{\text{exp}}$, where our method effectively applies no regularization, the bound trivially holds with $\delta = 0$. Leveraging this assumption, we present the following theorem, which establishes a stronger lower bound on policy improvement than the one in the original TRPO analysis (Schulman et al., 2015).

**Theorem 1** (Enhanced Monotonic Improvement). *Let $\alpha_1 = \max_s D_{KL}(\pi(\cdot|s)\|\pi_{old}(\cdot|s))$ and $\alpha_2 = \max_s D_f(\pi(\cdot|s)\|\pi_{pef}(\cdot|s))$, where the divergence $D_f$ can be the forward-KL, $\alpha$-divergence, or Jensen-Shannon divergence. If Assumption 1 holds, then the following bound on policy improvement is guaranteed:*

$$J(\pi) - L_{\pi_{old}}(\pi) \geq -\frac{2\gamma\alpha_1\epsilon_\pi}{(1-\gamma)^2} + \epsilon_f \tag{8}$$

*where $\epsilon_f = \frac{\delta}{1-\gamma} - \frac{C_f\gamma\alpha_2\epsilon_{pef}}{(1-\gamma)^2}$, $\epsilon_\pi = \max_{s,a}|A_\pi(s,a)|$ and $\epsilon_{pef} = \max_{s,a}|A_{\pi_{pef}}(s,a)|$, $C_f$ is a positive constant depending on the choice of the f-divergence.*

Comparing with the monotonic improvement theorem in Schulman et al. (2015), where the right hand side of the inequality is $-\frac{2\gamma\alpha_1\epsilon_\pi}{(1-\gamma)^2}$, we have an additional term $\epsilon_f$. On the distribution associated with expert dataset $\mathcal{D}_{\text{pef}}$, where $\delta > 0$ and effective regularization keeps $\alpha_2$ small, $\epsilon_f$ becomes a positive bonus. This results in a strictly better lower bound on policy improvement than in the original TRPO analysis, rigorously justifying that our method accelerates convergence by leveraging known expert behavior. On the exploration set $\mathcal{D}_{\text{exp}}$, the bound gracefully reduces to the standard TRPO guarantee. The proof of Theorem 1 follows a similar line of reasoning to the analyses in Schulman et al. (2015) and Kang et al. (2018). The detailed derivation is deferred to the Appendix J for the completeness.

## 5 EXPERIMENTS

In this section, we conduct extensive experiments to demonstrate the effectiveness and generalization of DPH-RL. The experimental setups were applied to two types of tasks: SQL (where the LLM generates SQL code and executes it using a tool) and mathematical reasoning, using both Llama 3.1 Meta AI (2024) and Qwen2.5 Team (2024) series models. In our experiments, we partition the dataset into $\mathcal{D}_{\text{exp}}$ and $\mathcal{D}_{\text{pef}}$ based on the capabilities of each model. For the Llama model, we define the $\mathcal{D}_{\text{pef}}$ using a success rate of 6 out of 8 attempts, while for the Qwen model, this threshold is raised to 7 out of 8 attempts. Since our method is orthogonal to existing GRPO variants, we use GRPO, DAPO, and reverse-KL (RKL) as baselines. Please refer to Appendix G for training details.

### 5.1 SQL

As shown in Table 1, in the Bird dataset, the Pass@8 scores for both GRPO and DAPO are lower than the base model, while our DPH-F and DPH-JS methods surpass the base model. This indicates that our strategy possesses a more robust ability to maintain model diversity. Specifically, DPH-JS shows Pass@8 scores that are 4.3% and 3.3% higher than GRPO and DAPO, respectively. For the Spider dataset, which is a cross-domain SQL task, the Pass@k metrics for all models generally suffer from performance degradation. However, DPH-F and DPH-JS can maintain accuracy levels closer to the base model. While DAPO performs better than GRPO on the Bird dataset, its performance is more unstable on the cross-domain data. Regarding cross-domain performance preservation, DPH-F demonstrates a more powerful capability, with its Pass@16 scores being 9.0% higher than DAPO. This suggests that for simple tasks that the model can already handle correctly, preservation of its original capabilities is more crucial than pure exploration. We conducted experiments on **OmniSQL-32B**, with the results presented in Table 8. Its Greedy performance on the Bird dataset surpasses all current open-source models.

Table 1: Results of **Llama-3.1-8B-Instruct** on SQL Tasks. The highest scores for RL Models are highlighted . Figure 6 shows the error bars for each method over three training runs.

| Model | Bird | | | Spider | | |
|---|---|---|---|---|---|---|
| | Greedy | Pass@8 | Pass@16 | Greedy | Pass@8 | Pass@16 |
| Base Model | 42.4 | 68.8 | 75.0 | 71.0 | 90.9 | 93.2 |
| *RL Models* | | | | | | |
| GRPO | 58.5 | 66.2 | 67.7 | 73.0 | 79.5 | 80.6 |
| DAPO | 60.0 | 67.2 | 69.0 | 71.1 | 75.3 | 76.7 |
| RKL | 60.0 | 69.8 | 71.8 | 71.0 | 79.0 | 80.6 |
| DPH-F | 60.4 | 70.1 | 71.6 | 77.4 | 84.5 | 85.7 |
| $\alpha$ Divergence | | | | | | |
| -$\alpha$=0.2 | 60.1 | 67.0 | 69.1 | 75.4 | 78.8 | 80.1 |
| -$\alpha$=0.5 | 60.8 | 68.0 | 69.8 | 75.2 | 80.4 | 82.5 |
| -$\alpha$=0.8 | 60.5 | 69.2 | 70.4 | 75.8 | 81.7 | 83.6 |
| DPH-JS (*Generator*) | | | | | | |
| -$\eta$=0.01 | 59.8 | 66.7 | 68.1 | 76.4 | 79.1 | 80.8 |
| -$\eta$=0.05 | 61.3 | 69.5 | 70.9 | 75.3 | 82.7 | 83.8 |
| -$\eta$=0.2 | 62.8 | 70.5 | 72.4 | 76.0 | 82.7 | 84.1 |
| DPH-JS (*Divergence Definition*) | | | | | | |
| -$\eta$=0.01 | 59.6 | 66.6 | 68.2 | 76.7 | 80.5 | 81.5 |
| -$\eta$=0.05 | 61.4 | 69.6 | 71.3 | 75.9 | 83.5 | 84.7 |
| -$\eta$=0.2 | 62.4 | 70.1 | 71.9 | 78.6 | 85.2 | 86.7 |

Table 2: Out-of-Domain Performance. Evaluating SQL-trained models on mathematical tasks. $\alpha = 0.2$ indicates the use of $\alpha$-divergence with $\alpha = 0.2$.

| Model | Pass@64 | | Pass@16 | | | Pass@8 | Avg |
|---|---|---|---|---|---|---|---|
| | AIME24 | AMC23 | Math500 | Olympiad | Minerva | College Math | |
| **Llama-3.1-8B-Instruct (SQL-trained)** | | | | | | | |
| Base Model | 40.0 | 95.0 | 81.2 | 46.4 | 54.0 | 45.5 | 60.35 |
| *RL Models* | | | | | | | |
| -GRPO | 33.3 | 77.5 | 72.0 | 41.8 | 51.2 | 38.4 | 52.37 |
| -DAPO | 30.0 | 77.5 | 72.8 | 44.4 | 52.3 | 38.8 | 52.63 |
| -RKL | 23.3 | 72.5 | 70.8 | 33.8 | 49.2 | 41.1 | 48.45 |
| -$\alpha$=0.2 | 40.0 | 90.0 | 79.8 | 44.0 | 51.8 | 41.8 | 57.90 |
| -$\alpha$=0.5 | 36.7 | 85.0 | 80.2 | 43.7 | 50.7 | 43.4 | 56.62 |
| -$\alpha$=0.8 | 43.3 | 87.5 | 80.8 | 43.1 | 54.0 | 44.2 | 58.82 |
| -DPH-F | 46.7 | 95.0 | 80.8 | 46.1 | 54.3 | 43.0 | 60.98 |
| -DPH-JS | 40.0 | 92.5 | 81.2 | 48.2 | 53.8 | 45.7 | 60.23 |

**Ablation Study** Our method introduces a plug-and-play $f$-divergence loss, weighted by a hyperparameter $\eta$, which we analyze in an ablation study. As shown in Table 1, when the value of $\eta$ is minimal, its actual setup and performance approximate that of DAPO trained exclusively on $\mathcal{D}_{\text{pef}}$. Increasing the value of $\eta$ leads to steady growth in the Pass@16 score, demonstrating the effectiveness of our method. Additionally, we compared two implementations of $f$-divergence: the "Generator" form and the "Divergence Definition" form (detailed in Appendix F). Although both yield similar performance, the "Divergence Definition" form is computationally expensive as it requires resampling from $\pi_\theta$ and an additional reference model to calculate $\pi_{ref}(a|q)$. Consequently, the "Generator" form is significantly more time-efficient.

**Performance on OOD Tasks** As shown in Table 2, we evaluated the diversity of our RL-trained models on five out-of-domain (OOD) mathematical datasets using a Llama model. (We excluded the AIME25 dataset due to its high fluctuation, ensuring a fair comparison.) Without incorporating

additional general-purpose data, methods like GRPO and DAPO cause models to overfit to the SQL training domain, leading to a significant drop in performance on other OOD tasks. In contrast, our DPH-F and DPH-JS methods maintain a higher average performance, surpassing DAPO by 8.35% and 7.6%, respectively.

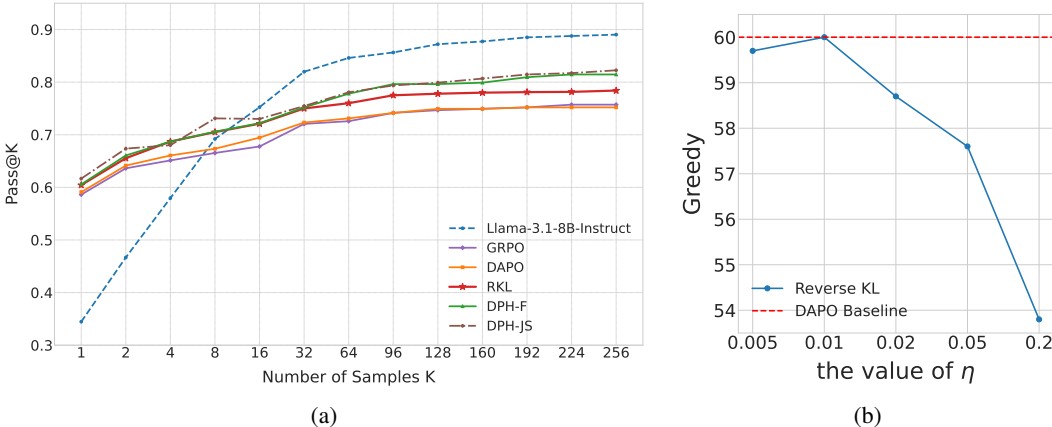

(a)                                                                (b)

Figure 2: Results of Llama models on the BIRD dataset. (a) Performance of different methods at different pass@k levels. (b) The impact of different KL term parameters on RKL.

**Performance with Massive Sampling**  We evaluated the performance of various methods as the number of samples increases. As illustrated in Figure 2a, the Pass@$k$ scores for all methods reach convergence by $k = 256$. Notably, standard GRPO and DAPO saturate early, plateauing at approximately 75%. Although DAPO outperforms GRPO at small $k$ values, its mechanism fails to scale effectively to larger $k$. The DPH family of methods significantly outperforms the GRPO baseline. Regarding Reverse KL, while it yields higher performance than GRPO, it remains significantly inferior to the DPH series. Overall, DPH-JS emerges as the superior method across both Pass@1 and Pass@$k$ metrics.

**Comparisons Under Different $f$-divergences**  In Tables 1 and 2, we analyzed the impact of different $f$-divergences on the Llama model. First, DAPO and GRPO, which lack any form of $f$-divergence constraint, experienced severe Pass@k collapse on both in-domain and out-of-domain tasks. Conversely, while RKL could maintain a high Pass@k on the training task's test set, its performance collapse on tasks with different distributions was even more severe than methods without any KL penalty. This highlights the limitations of reverse-KL: it causes the model to over-focus on the training data's distribution, thereby completely sacrificing generalization. To validate the reliability of our experimental baseline, we show the performance of RKL with different $\eta$ values in Figure 2b. When $\eta > 0.02$, the model's learning performance fails to surpass that of DAPO. This indicates that the chosen $\eta$ value is too large, and therefore our selection of $\eta = 0.01$ is well-justified. The $\alpha$ divergence sits between the forward-KL and reverse-KL, effectively preventing the model from veering toward extremes. Across all datasets, it shows a clear trend: as the $\alpha$ value increases, it theoretically approaches the capabilities of forward-KL, while experimentally maintaining a higher Pass@k. In contrast, DPH-F and DPH-JS methods demonstrated strong generalization across different tasks. The DPH-JS method, in particular, not only maintained a higher greedy performance on training tasks but also preserved a high Pass@k value on out-of-domain tasks, which fully demonstrates its significant advantage in preserving a multi-peak distribution.

## 5.2 MATHEMATICAL REASONING

For the mathematical tasks, we present two types of results, shown in Table 3 and Table 9, respectively. Table 3 shows the Pass@k scores on various test sets, while Table 9 shows the Mean@k scores, which is the average accuracy calculated by sampling the entire dataset k times. Regarding mathematical ability, different model families have completely different capabilities. For the Llama model, the improvement through RL is very limited; the average mean@k value when using GRPO only increased by 0.93, while Pass@k decreased by 3.26. By contrast, the Qwen model is

Table 3: The Pass@k metric for models trained on math datasets. In the Llama experiments, the DAPO settings ($\epsilon_{high}$=0.28) caused training to crash. To conduct effective experiments, we aligned the settings for all Llama experiments on the Math dataset with GRPO ($\epsilon_{high}$=0.2).

| Model | Pass@64 | | | Pass@16 | | | Pass@8 | Avg |
|---|---|---|---|---|---|---|---|---|
| | AIME24 | AIME25 | AMC23 | Math500 | Olympiad | Minerva | College Math | |
| **Llama-3.1-8B-Instruct** | | | | | | | | |
| Base Model | 40.0 | 23.3 | 95.0 | 81.2 | 46.4 | 54.0 | 45.5 | 55.06 |
| RL Models | | | | | | | | |
| -GRPO | 33.3 | 26.7 | 80.0 | 79.6 | 43.3 | 56.6 | 43.1 | 51.80 |
| -RKL | 36.7 | 16.7 | 75.0 | 80.0 | 39.1 | 56.6 | 43.4 | 49.64 |
| -DPH-F | 36.7 | 26.7 | 90.0 | 80.6 | 44.3 | 57.3 | 45.5 | 54.44 |
| -DPH-JS | 40.0 | 26.7 | 82.5 | 81.4 | 45.8 | 58.1 | 45.1 | 54.23 |
| **Qwen2.5-Math-7B** | | | | | | | | |
| Base Model | 63.3 | 56.7 | 87.5 | 88.8 | 61.9 | 56.6 | 42.9 | 65.39 |
| RL Models | | | | | | | | |
| -GRPO | 56.6 | 50.0 | 97.5 | 93.0 | 62.8 | 64.0 | 50.1 | 67.71 |
| -DAPO | 63.3 | 46.7 | 97.5 | 92.2 | 63.1 | 64.3 | 48.7 | 67.97 |
| -RKL | 66.7 | 40.0 | 97.5 | 92.0 | 64.6 | 64.0 | 51.3 | 68.01 |
| -DPH-F | 73.33 | 50.0 | 97.5 | 92.4 | 63.8 | 64.8 | 50.9 | 70.39 |
| -DPH-JS | 66.7 | 53.3 | 100.0 | 92.8 | 65.2 | 66.2 | 51.0 | 70.74 |

a more capable model, which allows it to achieve a significant boost in both Mean@k (approximately 20%) and Pass@k. We selected these two models to explore the performance of the DPH method under different model strengths. For Llama, DPH-JS maintained the original Pass@k value and significantly improved mean@k on AIME, which demonstrates that DPH-JS is a powerful and highly versatile method with strong capabilities in both exploration and preservation. Furthermore, DPH-JS provided a more balanced improvement than GRPO, increasing both Pass@k and mean@k averages, showing it does not sacrifice one metric for the other. For the Qwen model, the overall performance showed a dual trend: a slight decrease in Pass@k on the difficult AIME test set, but an increase on other, simpler datasets. In both scenarios, the DPH-JS method achieved the best performance. On AIME, it maintained a diversity closer to that of the base model compared to both GRPO and DAPO. On the other datasets, it achieved a higher Pass@k. This trend highlights a key challenge for large models in RL: the need to both solidify existing knowledge while also exploring new boundaries. The DPH-JS method successfully balances these two conflicting goals.

# 6 ANALYSIS

## 6.1 ANALYSIS OF KEEP AND EXPLORATION

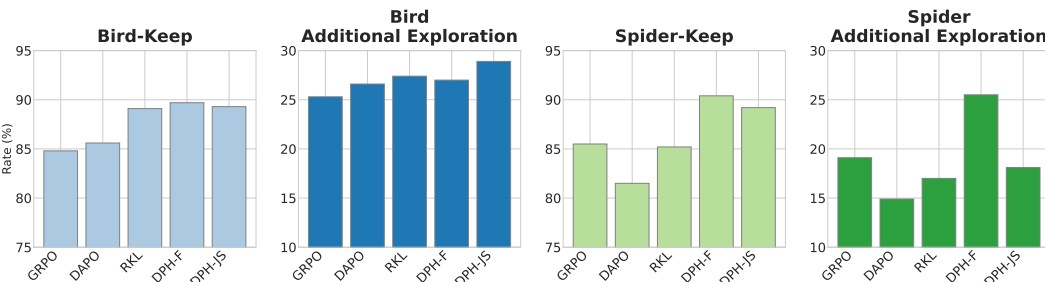

Figure 3: Evaluating RL-Tuned Llama models vs. base model: knowledge retention (keep) vs. new solution discovery (additional exploration).

To gain a more granular understanding of why our method achieves higher Pass@k scores, we extracted correct and incorrect samples from the base model's Pass@8 results. These were designated as data subsets $D_{sub}^{correct}$ and $D_{sub}^{wrong}$, respectively. We then compared them with the model after

reinforcement learning. The proportion of samples it got correct on $D_{sub}^{correct}$ is its keep rate, while the proportion it got correct on the incorrect samples from $D_{sub}^{wrong}$ is its additional exploration rate. As shown in Figure 3, the keep rate of GRPO and DAPO both decreased on the two datasets, while the KL method mainly maintained a higher keep rate to maintain a higher Pass@k value. DAPO had higher exploration within the domain than GRPO, but the keep rate outside the domain dropped significantly. RKL could only have a high keep rate on the same Bird test set as the training set, while it dropped to a level similar to GRPO on Spider. DPH-JS had higher exploration on the Bird test set, while DPH-F was higher on the Spider dataset. Furthermore, Appendix H.2 shows that DPH is the most stable and effective method during training, while other mainstream approaches either have poor generalization (e.g., DAPO) or diversity collapse late in training (e.g., GRPO and RKL).

## 6.2 IMPACT OF SPLITTING $\mathcal{D}_{\text{EXP}}$ AND $\mathcal{D}_{\text{PEF}}$

Table 4: '8 of 8' indicates a threshold where samples correctly solved in 8 out of 8 attempts are assigned to $\mathcal{D}_{\text{pef}}$ in DPH-JS. 'Full Data' implies using the entire dataset for both $\mathcal{D}_{\text{pef}}$ and $\mathcal{D}_{\text{exp}}$.

| Model | Size of $\mathcal{D}_{\text{exp}}$ | Size of $\mathcal{D}_{\text{pef}}$ | Bird | | | Spider | | |
|---|---|---|---|---|---|---|---|---|
| | | | Greedy | Pass@8 | Pass@16 | Greedy | Pass@8 | Pass@16 |
| GRPO | 8958 | 0 | 58.5 | 66.2 | 67.7 | 73.0 | 79.5 | 80.6 |
| DAPO | 8958 | 0 | 60.0 | 67.2 | 69.0 | 71.1 | 75.3 | 76.7 |
| 8 of 8 | 8266 | 692 | 61.4 | 71.6 | 73.5 | 75.4 | 86.3 | 87.2 |
| 7 of 8 | 7429 | 1529 | 61.3 | 71.2 | 73.6 | 75.1 | 85.8 | 87.2 |
| 6 of 8 | 6710 | 2248 | 62.8 | 70.5 | 72.4 | 76.0 | 82.7 | 84.1 |
| 5 of 8 | 5783 | 3175 | 60.2 | 70.6 | 74.0 | 74.3 | 86.2 | 88.9 |
| Full Data | 8958 | 8958 | 57.4 | 73.1 | 76.6 | 71.8 | 88.4 | 90.3 |

In the main experiments, we established an accuracy threshold to partition the data into $\mathcal{D}_{\text{exp}}$ and $\mathcal{D}_{\text{pef}}$. This section provides a detailed analysis of the impact of this threshold. As illustrated in Table 4, the performance difference between the '7 of 8' and '8 of 8' settings is negligible. At '6 of 8', the model sacrifices some diversity to achieve higher greedy performance; by removing simpler samples from $\mathcal{D}_{\text{exp}}$, the model focuses exploration on more challenging problems, while the KL term ensures continued learning on well-mastered samples. Conversely, at '5 of 8', the significant reduction in $\mathcal{D}_{\text{exp}}$ data leads to a decline in greedy performance, despite yielding improved Pass@$k$ scores. In the 'Full Data' setting, greedy performance is poor—even inferior to the GRPO baseline—yet it achieves the highest Pass@$k$. Although the large volume of $\mathcal{D}_{\text{pef}}$ preserves sufficient diversity, 'Full Data' suffers from two critical issues: (1) Our goal is to preserve the diversity of correct solutions, not the noise of incorrect ones. Forcing the selection of a single correct response from samples solved correctly only once out of eight attempts introduces statistical bias. (2) Regarding the training objective, the model can trivially mimic the sole correct sample for each query to maximize reward and minimize the KL term, thereby optimizing the loss without genuine learning. In contrast, '8 of 8' represents an unbiased statistical approach requiring no artificial sampling. To compensate for the reduced data scale in this setting, we retain all 8 sampled trajectories for each instance during training. In our experiments, although '8 of 8' allocates only 8% of the training data to $\mathcal{D}_{\text{pef}}$, it comprehensively outperforms DAPO and GRPO. This demonstrates the robustness of DPH-RL even under extreme conditions, suggesting that directly adopting the '8 of 8' threshold is the most straightforward strategy for its application.

## 7 CONCLUSION

We propose DPH-RL to overcome the limitations of reverse KL divergence in reinforcement learning for LLMs. By leveraging $f$-divergence, our method mitigates both diversity collapse and catastrophic forgetting. We demonstrate that two implementations—a "Generator" form that requires no reference model during training and a "Divergence Definition" form—yield similar performance. This makes our method highly adaptable, providing enhanced capabilities with computational costs comparable to GRPO. We explicitly recommend the DPH-JS (Generator) variant with the '8 of 8' threshold as the optimal configuration for future researchers. However, a key limitation, shared with GRPO, is the dependence on the base model's capabilities, which become scarce on more challenging datasets. Future work will therefore focus on resolving this trade-off, aiming to develop methods that can both learn effectively from imperfect data and mitigate the resulting estimation bias.

## REPRODUCIBILITY

Our datasets are all based on open-source datasets. The experimental methodology, data proportions, and hyperparameter settings are detailed in Appendix G.

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

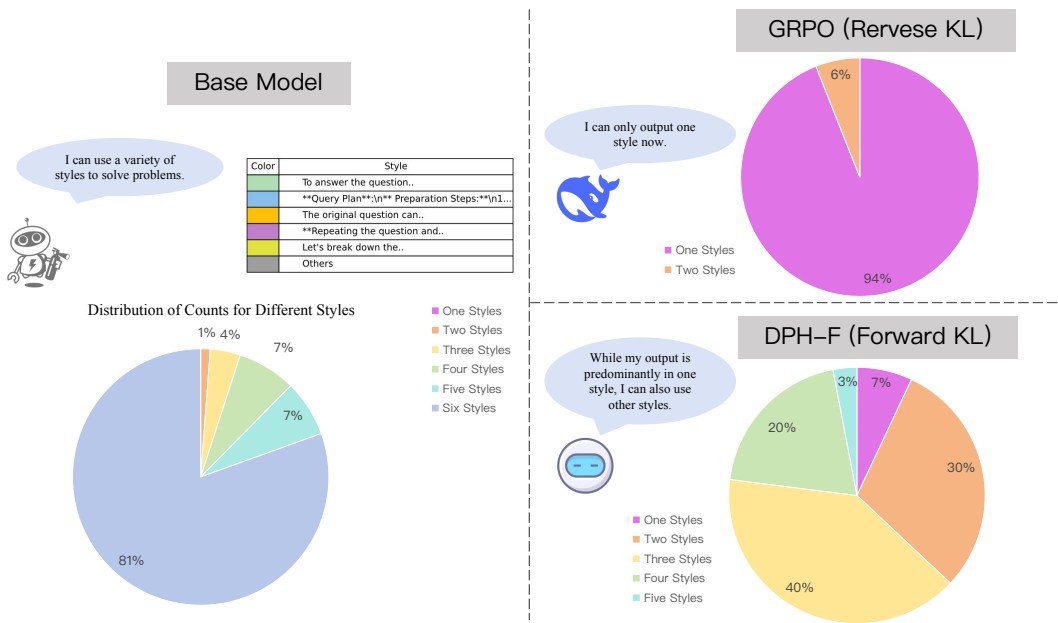

Figure 4: On the left, we construct a base model that outputs multiple solution styles for SQL problems. This model is then used for reinforcement learning training. We calculated the probability of the number of times the model outputted different styles across 32 samples.

## A    THE USE OF LARGE LANGUAGE MODELS

In our work, we exclusively use LLMs for writing refinement, which means we first write a piece of text ourselves, and then use the LLM to correct grammar, formatting, and other issues. For our experiments, we also use LLMs to help us fix code bugs and generate Python code for plotting.

## B    MULTIPLE STYLE CAPABILITY EXPERIMENT

We construct a base model that outputs five different solution styles for SQL problems. This model is then used for reinforcement learning training. Next, we sample each question 32 times. Based on the output prefix, we count the number of times each distinct style appears within these 32 samples. For prefixes that are not among our original five styles (e.g., the model combines several different prefixes), we categorize them as "Others" and treat them as an additional style.

The result shows in Figure 4. On the base model, we find that it can fully output all styles. However, after training with reverse KL-constrained GRPO, most outputs degenerate into a single style. In contrast, forward KL significantly mitigates the degradation of the model's ability to produce different styles.

## C    PRELIMINARIES ON $f$-DIVERGENCE

In Table 5, we summarize several typical $f$-divergences and their corresponding generator functions $f(u)$, with the definitions of forward and reverse KL divergence adapted according to the latest research. All listed divergences adhere to the standard properties of the generator function: strict convexity, lower semi-continuity, and the normalization condition $f(1) = 0$. Notably, this table clarifies that both forward and reverse KL divergence can be unified as special cases of the $\alpha$-divergence under specific parameters. This provides a unified mathematical framework for measuring and optimizing distribution shifts using $\pi_{\text{ref}}$ in subsequent model training.

Table 5: Summary of some typical f-divergences $D_f(p\|q)$ together with their generator functions $f(u)$. In this paper, we let $\pi_{ref}$ be $q$, and the formula for f-divergence references an improved version (Wang et al., 2024), the definitions for forward-KL and reverse-KL are swapped compared to Wikipedia. For all divergences, the generator function $f: \mathbb{R}^+ \to \mathbb{R} \cup \{+\infty\}$ is strictly convex, lower-semicontinuous, and satisfies $f(1) = 0$. Forward-KL and reverse-KL are the special case of alpha divergence.

| Name | Divergence Definition $D_f(p\|q)$ | Generator $f(u)$ |
|---|---|---|
| Reverse-KL($\alpha = 0$) | $\int p(x) \log \frac{p(x)}{q(x)} dx$ | $u \log u$ |
| Forward-KL($\alpha = 1$) | $\int q(x) \log \frac{q(x)}{p(x)} dx$ | $-\log u$ |
| $\alpha$-divergence ($\alpha \notin \{0,1\}$) | $\frac{1}{\alpha(\alpha-1)} \int q(x) \left[ \left( \frac{p(x)}{q(x)} \right)^\alpha \right.$ $\left. -\alpha \left( \frac{p(x)}{q(x)} \right) - 1 \right] dx$ | $(u^{1-\alpha} - (1-\alpha)u - \alpha)/(\alpha(\alpha-1))$ |
| Jensen-Shannon | $\frac{1}{2} \int \left( p(x) \log \frac{2p(x)}{p(x)+q(x)} \right.$ $\left. +q(x) \log \frac{2q(x)}{p(x)+q(x)} \right) dx$ | $\frac{u}{2} \log u - \frac{(u+1)}{2} \log \frac{u+1}{2}$ |

# D  THEORETICAL CONNECTION TO EXPERIENCE REPLAY AND REGULARIZED REHEARSAL

In this section, we elaborate on the theoretical interpretation of DPH-RL as a hybrid method rooted in replay mechanisms. We demonstrate that while our method incorporates a replay buffer component, it significantly extends standard Experience Replay (ER) through the introduction of symmetric divergences.

## D.1  DPH-F AS DIVERGENCE-BASED EXPERIENCE REPLAY

Minimizing the Forward-KL divergence, defined as $\mathbb{E}_{x \sim \pi_{ref}}[-\log \pi_\theta(x)]$, is mathematically equivalent to maximizing the log-likelihood of trajectories sampled from the reference policy $\pi_{ref}$. In our framework, $\mathcal{D}_{pef}$ serves as a curated buffer containing "perfect" trajectories. Consequently, the DPH-F objective acts effectively as an Experience Replay mechanism. By enforcing probability mass coverage on these high-quality solutions, it prevents the model from catastrophically forgetting previously mastered skills while optimizing rewards on the exploration set $\mathcal{D}_{exp}$.

## D.2  DPH-JS AS REGULARIZED REHEARSAL VIA DUAL-SIDED SAMPLING

While DPH-F acts as direct rehearsal, DPH-JS (Jensen-Shannon) introduces a more nuanced mechanism. Since the JS divergence is symmetric and defined based on a mixture distribution $M = (\pi_{ref} + \pi_\theta)/2$, its optimization necessitates **dual-sided sampling**.

Unlike standard replay which only samples from the historical buffer, minimizing DPH-JS involves comparing samples from two sources:

- **The Reference Buffer** ($\pi_{ref}$): Anchoring the model to the valid distribution to prevent collapse.
- **The Current Policy** ($\pi_\theta$): Sampling self-generated trajectories to regulate the model's deviation from the mixture mean.

This dual dependency makes DPH-JS a form of **"Regularized Rehearsal."** It anchors the model to the reference distribution (similar to ER) to ensure diversity, but simultaneously regulates the model's own exploration. This prevents the model from drifting too far into unstable regions without forcing it to strictly "memorize" the buffer as rigidly as Forward-KL or SFT might.

# E  RLVR ALGORITHMS

## E.1  GROUP RELATIVE POLICY OPTIMIZATION (GRPO)

GRPO presents an innovative approach to policy learning that distinguishes itself from methods like Proximal Policy Optimization (PPO) by eliminating the need for an explicit value function. Instead, GRPO computes the advantage in a group-relative manner, offering a streamlined yet effective optimization strategy.

For a specific question-answer pair $(q, a)$, GRPO's underlying behavior policy, $\pi_{\theta_{\text{old}}}$, generates a group of $G$ individual responses, denoted as $\{o_i\}_{i=1}^G$. The advantage for the $i$-th response within this ensemble is then precisely calculated by normalizing the rewards specific to that group, $\{R_i\}_{i=1}^G$:

$$\hat{A}_{i,t} = \frac{r_i - \text{mean}(\{R_i\}_{i=1}^G)}{\text{std}(\{R_i\}_{i=1}^G)}. \tag{9}$$

Similar to the clipped surrogate objective found in PPO, GRPO also incorporates a clipping mechanism to constrain policy updates. This helps maintain training stability and improve sample efficiency by ensuring that new policies don't deviate too drastically from previous ones. Beyond this, GRPO further enhances regularization by directly adding a Kullback-Leibler (KL) divergence penalty term to its objective function. This penalty helps prevent the policy from drifting too far from a reference policy, promoting stable and controlled learning.

The comprehensive objective function for GRPO is articulated as:

$$\mathcal{L}\text{GRPO}(\theta) = -\mathbb{E}_{\substack{(q,a)\sim\mathcal{D} \\ o_i\sim\pi_{\theta_{\text{old}}}(\cdot|q)}} \left[ \frac{1}{G}\sum_{i=1}^G \frac{1}{|o_i|}\sum_{t=1}^{|o_i|} \Bigg( \right.$$
$$\min\left(\rho_{i,t}(\theta)\hat{A}_{i,t}, \ \text{clip}\left(\rho_{i,t}(\theta), 1-\varepsilon, 1+\varepsilon\right)\hat{A}_{i,t}\right) \tag{10}$$
$$\left. - \eta D_{\text{KL}}(\pi_\theta||\pi_{\text{ref}}) \Bigg) \right],$$

where $\eta$ is a coefficient for the KL penalty, $D_{\text{KL}}(\pi_\theta||\pi_{\text{ref}})$ quantifies the KL divergence between the current policy $\pi_\theta$ and a specified reference policy $\pi_{\text{ref}}$. The term $\rho_{i,t}(\theta)$ represents the importance sampling ratio for the $i$-th response at time $t$, which is defined as:

$$\rho_{i,t}(\theta) = \frac{\pi_\theta(o_{i,t} \mid q, o_{i,<t})}{\pi_{\theta_{\text{old}}}(o_{i,t} \mid q, o_{i,<t})}. \tag{11}$$

A critical design choice in GRPO is its sample-level objective computation. Specifically, the loss is first averaged within each generated sequence, and subsequently, these sequence-level losses are averaged across various samples. This distinct computational approach, particularly when compared to token-level optimizations, can significantly influence the algorithm's empirical performance.

Traditional GRPO applies a single, unified loss function to all training samples. This loss includes a KL divergence penalty, typically in the form of reverse KL divergence, which aims to keep the fine-tuned policy close to the base policy. This penalty is added directly to the policy gradient loss. When applied to our experimental setup, the KL penalty for samples from $\mathcal{D}_{\text{exp}}$ would be:

$$\mathcal{L}_{\text{KL-GRPO}} = \mathbb{E}_{q\sim\mathcal{D}_{\text{exp}}}\left[-\beta D_{\text{KL}}(\pi_\theta||\pi_{\text{base}})\right], \tag{12}$$

where the reverse KL divergence is defined as:

$$D_{\text{KL}}(\pi_\theta||\pi_{\text{base}}) = \sum_x \pi_\theta(x)\log\left(\frac{\pi_\theta(x)}{\pi_{\text{base}}(x)}\right). \tag{13}$$

Here, the term $D_{\text{KL}}(\pi_\theta||\pi_{\text{base}})$ represents the **Reverse KL Divergence**. It measures the information lost when the base policy $\pi_{\text{base}}$ approximates the new policy $\pi_\theta$. This choice heavily penalizes the

new policy $\pi_\theta$ for exploring actions that the base policy $\pi_{\text{base}}$ considers to have a low probability. In other words, if the new policy $\pi_\theta$ assigns a high probability to an action where the base policy $\pi_{\text{base}}$ assigns a near-zero probability, this divergence will be very large. The effect is to strongly encourage the new policy $\pi_\theta$ to stick to the modes (high-probability regions) of the original base policy $\pi_{\text{base}}$, which restricts the exploration space.

### E.2 DYNAMIC SAMPLING POLICY OPTIMIZATION (DAPO)

DAPO is an enhancement of the GRPO algorithm, incorporating several key improvements. DAPO eliminates the KL penalty and refines the clipping mechanism, changing the upper bound from $(1 + \varepsilon)$ to a fixed value of $(1 + \varepsilon_{\text{upper}})$, where $\varepsilon_{\text{upper}}$ is set to 0.28. A core innovation of DAPO is its dynamic sampling mechanism, which moves beyond the "all-or-nothing" approach to sampling. Additionally, the algorithm applies a token-level policy gradient loss and uses an overlong reward shaping technique.

## F METHOD FOR DIVERGENCE DEFINITION

For f-divergences like forward KL, the Divergence Definition and Generator implementations are equivalent. However, for divergences such as JS divergence, which require sampling from both the reference policy $\pi_{ref}$ and the new policy $\pi_\theta$:

$$\text{JS}(\pi_{\text{base}}||\pi_\theta) = \frac{1}{2}D_{\text{KL}}\left(\pi_{\text{base}}||\frac{\pi_{\text{base}} + \pi_\theta}{2}\right) + \frac{1}{2}D_{\text{KL}}\left(\pi_\theta||\frac{\pi_{\text{base}} + \pi_\theta}{2}\right) \qquad (14)$$

Since the JS divergence is composed of two parts, we use data from the $\mathcal{D}_{\text{pef}}$ loss to compute the first part $D_{\text{KL}}\left(\pi_{\text{base}}||\frac{\pi_{\text{base}} + \pi_\theta}{2}\right)$. For the training on the $\mathcal{D}_{\text{exp}}$ loss, we introduce a reference model to calculate the value of the second part $D_{\text{KL}}\left(\pi_\theta||\frac{\pi_{\text{base}} + \pi_\theta}{2}\right)$.

The loss for DPH-JS on the $\mathcal{D}_{\text{exp}}$ can be written as:

$$\mathcal{L}_{\text{DPH-exp}}(\theta) = -\mathbb{E}_{\substack{(q,a)\sim\mathcal{D}_{\text{exp}} \\ o_i\sim\pi_{\theta_{\text{old}}}(\cdot|q)}}\left[\frac{1}{G}\sum_{i=1}^{G}\frac{1}{|o_i|}\sum_{t=1}^{|o_i|}\min\left(\rho_{i,t}(\theta)\hat{A}_{i,t},\ \text{clip}\left(\rho_{i,t}(\theta), 1-\varepsilon, 1+\varepsilon\right)\hat{A}_{i,t}\right)\right.$$
$$\left. - \beta_1 D_{\text{KL}}\left(\pi_\theta||\frac{\pi_{\text{base}} + \pi_\theta}{2}\right)\right] \qquad (15)$$

By adjusting the value of $\beta_1$, we can make the values of the two JS components within the same batch as close as possible, thereby achieving balance during training. $\beta_1$ is not a hyperparameter; it only depends on the dataset size and $\eta$. Its formula is:

$$\beta_1 = \eta * \frac{|\mathcal{D}_{\text{pef}}|}{|\mathcal{D}_{\text{exp}}|} \qquad (16)$$

The loss for DPH-JS on the $\mathcal{D}_{\text{pef}}$ can be expressed as:

$$\mathcal{L}_{\text{DPH-pef}} = \mathbb{E}_{q\sim\mathcal{D}_{\text{pef}}}\left[D_{\text{KL}}\left(\pi_{\text{base}}||\frac{\pi_{\text{base}} + \pi_\theta}{2}\right)\right] \qquad (17)$$

## G TRAINING DETAILS

### G.1 TASK SETTINGS

For the SQL task, we used the Llama-3.1-8B-Instruct and OmniSQL32B models. We performed RL training exclusively on the BIRD dataset Li et al. (2024b) and then evaluated the models on both the BIRD and Spider (Cross-Domain) Yu et al. (2018) datasets. To validate the generalization of our method, we also tested it on a mathematical reasoning task, treating it as an out-of-domain (OOD) evaluation.

For the mathematical reasoning task, we used the Llama-3.1-8B-Instruct and Qwen2.5-Math-7B models. We conducted RL training on a DAPO-17K subset, having first filtered out problems that initially had a Pass@8 score of 0. For testing, we used seven different datasets: AIME24 Li et al. (2024a), AIME25, AMC23 Li et al. (2024a), Math500 Hendrycks et al. (2021), Olympiad He et al. (2024), Minerva Lewkowycz et al. (2022), and College Math Tang et al. (2024).

**Evaluation**  For the SQL tasks, the model will generate an SQL statement, and then we call the SQL executor to get the result. This result is typically a tuple, such as [(string, int), (string, int)]. We compute the Cartesian product of the prediction and the ground truth. For the math tasks, we use the official Qwen2.5 evaluation tool[1] for a detailed assessment, as this tool provides tailored evaluations for each dataset.

## G.2  METHOD SETTINGS

Table 6: Configuration Comparison of RL Methods

| Method | Online Data | Offline Data | Offline Loss | $\eta$ | $\epsilon_{high}$ | Dynamic Sampling |
|---|---|---|---|---|---|---|
| GRPO | $\mathcal{D}_{exp},\mathcal{D}_{pef}$ | - | - | - | 0.2 | × |
| DAPO | $\mathcal{D}_{exp},\mathcal{D}_{pef}$ | - | - | - | 0.28 | ✓ |
| RKL | $\mathcal{D}_{exp},\mathcal{D}_{pef}$ | - | - | 0.01 | 0.28 | ✓ |
| $\alpha$-divergence | $\mathcal{D}_{exp}$ | $\mathcal{D}_{pef}$ | $\alpha$-divergence | 0.01 | 0.28 | ✓ |
| DPH-F | $\mathcal{D}_{exp}$ | $\mathcal{D}_{pef}$ | Forward KL | 0.01 | 0.28 | ✓ |
| DPH-JS | $\mathcal{D}_{exp}$ | $\mathcal{D}_{pef}$ | JS divergence | 0.2 | 0.28 | ✓ |

The implementation of RKL is consistent with the standard GRPO implementation. Our default implementation for DPH-RL uses the **Generator** method. The settings for our different methods are detailed in Table 6. For all RL algorithms, we consistently use a token-level loss. In most of our experiments, the $\epsilon_{high}$ for our DPH methods was set to 0.28. However, due to training instability in the Llama math experiments, we set the $\epsilon_{high}$ to 0.2 for all methods.

## G.3  HYPERPARAMETERS

Table 7: Hyperparameters for RL Training

| Hyperparameter | Llama-SQL | OmniSQL-32B-SQL | Llama-Math | Qwen2.5-Math |
|---|---|---|---|---|
| Batch Size | 128 | 128 | 128 | 128 |
| Learning Rate | 1e-6 | 1e-6 | 2e-7 | 2e-7 |
| Rollout Temperature | 1.0 | 1.0 | 1.0 | 1.0 |
| Rollout Top-p | 0.95 | 0.95 | 0.95 | 0.95 |
| Validation Temperature | 1.0 | 1.0 | 0.6 | 0.6 |
| Validation Top-p | 0.95 | 0.95 | 0.95 | 0.95 |
| PPO Epochs | 1 | 1 | 1 | 1 |
| Size of $\mathcal{D}_{exp}$ | 6710 | 5169 | 4599 | 7009 |
| Size of $\mathcal{D}_{pef}$ | 2248 | 3789 | 1248 | 3199 |
| Max Response Length | 4096 | 4096 | 2048 | 2048 |
| Number of Rollouts | 16 | 16 | 16 | 16 |
| Training Epochs | 4 | 4 | 8 | 8 |

All experiments were conducted on 32 NVIDIA A800-80G GPUs using the VeRL framework for our RL algorithm implementations. For all experiments, we set the number of rollouts to 16. In our setup, the training batch size was 128, and the batch size for the $D_{pef}$ data was 256. This resulted in using 2048 samples from $D_{exp}$ and 256 samples from $D_{pef}$ at each learning step, yielding an effective ratio of 8:1. A comprehensive list of the specific hyperparameters used is provided in Table 7.

In the offline phase, we used the current model to perform 8 rollouts on the training data. For the SQL tasks, we separated the perfectly correct examples from the rollouts and used the remaining ones to construct $\mathcal{D}_{exp}$. For the DAPO-17k dataset used in the math tasks, we discarded all data where the model's responses were entirely incorrect.

---

[1] https://github.com/QwenLM/Qwen2.5-Math

# H ADDITIONAL EXPERIMENTS

## H.1 32B EXPERIMENTS FOR SQL

Table 8: Results on Omnisql-32B for SQL tasks. * Indicates that the results are from the Bird Bench.

| Model | Bird | | | Spider | | |
|---|---|---|---|---|---|---|
| | Greedy | Pass@8 | Pass@16 | Greedy | Pass@8 | Pass@16 |
| **OmniSQL-32B** | | | | | | |
| Base Model | 65.0 | 77.3 | 79.8 | 85.0 | 92.8 | 93.6 |
| *Other Models* | | | | | | |
| Infly-RL-SQL-32B* | 70.1 | - | - | - | - | - |
| XiYanSQL-QwenCoder-32B* | 67.0 | - | - | - | - | - |
| Arctic-ExCoT-70B* | 68.5 | | | | | |
| Command A* | 63.5 | | | | | |
| *RL Models* | | | | | | |
| GRPO | 69.4 | 76.3 | 78.8 | 84.7 | 91.6 | 92.4 |
| DAPO | 69.9 | 76.8 | 79.2 | 83.8 | 91.0 | 91.7 |
| DPH-F | 70.4 | 78.6 | 80.8 | 86.1 | 92.7 | 93.3 |
| DPH-JS | 70.5 | 79.2 | 81.9 | 84.9 | 91.9 | 93.1 |

We also tested the effectiveness of DPH-F and DPH-JS on a larger 32B model and found them to be equally effective. As shown in Table 8, please note that a greedy score of 70.5 on the Bird dataset is already approaching the performance limit of open-source Single-Models[2].

## H.2 ANALYSIS OF TRAINING PROGRESS

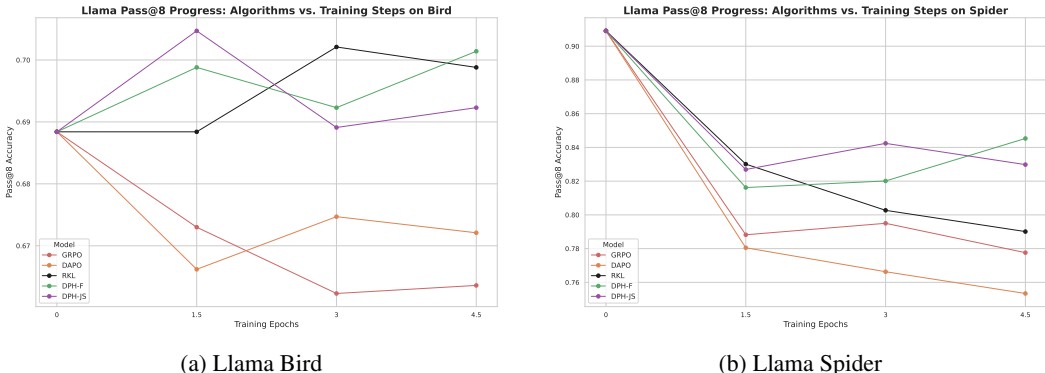

(a) Llama Bird                                     (b) Llama Spider

Figure 5: Llama Pass@8 progress: Algorithms vs. Training Steps

In Figure 5, we visualize the evolution of the Llama model's Pass@8 metric during training. On the Bird dataset, GRPO shows a clear, gradual collapse, while DAPO's performance continuously oscillates. Over the long term, DPH-F and DPH-JS both consistently maintain a higher Pass@8 value than the initial level. On the Spider dataset, DAPO performs extremely poorly, even falling below GRPO. Our experiments show that DAPO sacrifices out-of-domain generalization for improved performance on in-domain datasets. While RKL performs well on the Bird dataset, its performance also shows a gradual collapse on the Spider dataset as training progresses.

## H.3 ERROR BAR

## H.4 MEAN@K METRIC FOR MATHEMATICAL REASONING TASK

To verify that our reinforcement learning training was effective on the math tasks, we evaluated the performance using the mean@k metric in Table 9.

---

[2]https://bird-bench.github.io/

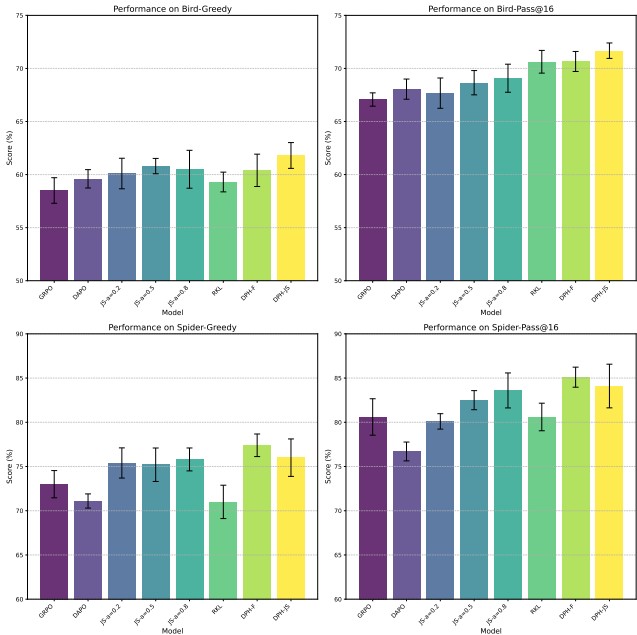

Figure 6: Error bar in Llama Sql. For each method in our Llama SQL experiments, we conducted three separate reinforcement learning training runs. We then selected the model that achieved the highest pass@16 score on the Bird.

Table 9: The Mean@k metric for models trained on math datasets. To accelerate training, we filtered the DAPO-17k dataset for data with a Pass@8 score of 0. In the end, we trained Llama on 5.8k samples and Qwen on 10.2k samples.

| Model | Mean@64 | | | Mean@16 | | | Mean@8 | Avg |
|---|---|---|---|---|---|---|---|---|
| | AIME24 | AIME25 | AMC23 | Math500 | Olympiad | Minerva | College Math | |
| **Llama-3.1-8B-Instruct** | | | | | | | | |
| Base Model | 5.5 | 0.5 | 23.0 | 46.5 | 15.2 | 22.1 | 27.3 | 20.01 |
| RL Models | | | | | | | | |
| -GRPO | 5.6 | 0.6 | 23.3 | 47.7 | 14.9 | 25.4 | 29.1 | 20.94 |
| -RKL | 6.4 | 0.2 | 22.3 | 45.6 | 13.0 | 24.8 | 28.6 | 20.13 |
| -DPH-F | 6.2 | 0.5 | 23.8 | 48.1 | 15.5 | 25.4 | 29.7 | 21.31 |
| -DPH-JS | 8.4 | 0.5 | 24.2 | 48.4 | 15.6 | 27.4 | 29.9 | 22.06 |
| **Qwen2.5-Math-7B** | | | | | | | | |
| Base Model | 12.7 | 6.6 | 32.2 | 48.9 | 22.8 | 13.3 | 19.3 | 22.26 |
| RL Models | | | | | | | | |
| -GRPO | 25.8 | 10.9 | 60.6 | 75.5 | 37.8 | 38.8 | 40.6 | 41.43 |
| -DAPO | 25.9 | 11.1 | 61.3 | 76.3 | 37.6 | 39.0 | 40.7 | 41.70 |
| -RKL | 25.8 | 10.3 | 61.5 | 75.3 | 36.7 | 34.6 | 39.9 | 40.59 |
| -DPH-F | 26.0 | 10.6 | 60.4 | 76.4 | 37.3 | 38.1 | 40.9 | 41.38 |
| -DPH-JS | 26.6 | 11.2 | 62.7 | 76.5 | 38.5 | 38.9 | 41.9 | 42.33 |

# I  CASE STUDY

We have listed the use cases and prompts for the two tasks in detail to facilitate future reproduction.

## I.1 SQL TASK

---

**Case Study-SQL**

System Prompt
You are Qwen, created by Alibaba Cloud. You are a helpful assistant.

User Prompt

Task Overview:
You are a data science expert. Below, you are provided with a database schema and a natural language question. Your task is to understand the schema and generate a valid SQL query to answer the question.
Database Engine:
SQLite
Database Schema:
CREATE TABLE frpm (
CDSCode text, – example: ['01100170109835', '01100170112607']
'Academic Year' text, – example: ['2014-2015']
'County Code' text, – example: ['01', '02']
'Free Meal Count (K-12)' real, – example: [565.0, 186.0]
'Percent (%) Eligible Free (K-12)' real, – example: [0.519779208831647, 0.470886075949367]
'2013-14 CALPADS Fall 1 Certification Status' integer, – example: [1]
PRIMARY KEY (CDSCode),
CONSTRAINT fk_frpm_cdscode FOREIGN KEY (CDSCode) REFERENCES schools (CDSCode)
);
CREATE TABLE....  This schema describes the database's structure, including tables, columns, primary keys, foreign keys, and any relevant relationships or constraints.
Question:
Eligible free rate for K-12 = 'Free Meal Count (K-12)' / 'Enrollment (K-12)'. What is the highest eligible free rate for K-12 students in the schools in Alameda County?
Instructions:
- Make sure you only output the information that is asked in the question. If the question asks for a specific column, make sure to only include that column in the SELECT clause, nothing more.
- The generated query should return all of the information asked in the question without any missing or extra information.
- Before generating the final SQL query, please think through the steps of how to write the query.

Output Format:
In your answer, please enclose the generated SQL query in a code block:

```sql
-- Your SQL query
```

Take a deep breath and think step by step to find the correct SQL query.

Response

To find the codes of the schools with a total enrollment of over 500, we need to follow these steps:
1. Identify the table that contains the enrollment data: The 'frpm' table contains the 'Enrollment (K-12)' and 'Enrollment (Ages 5-17)' columns, which represent the enrollment data.
2. Identify the relevant columns: We need to sum the 'Enrollment (K-12)' and 'Enrollment (Ages 5-17)' columns to get the total enrollment. 3. Identify the filter condition: We need to filter the schools with a total enrollment of over 500. 4. Write the SQL query: We can use a 'SELECT' statement to retrieve the 'CDSCode' column from the 'frpm' table, filtering the

---

schools where the sum of 'Enrollment (K-12)' and 'Enrollment (Ages 5-17)' is greater than 500.

Based on these steps, the SQL query to find the codes of the schools with a total enrollment of over 500 is:

```sql
SELECT frpm.CDSCode
FROM frpm
WHERE frpm.`Enrollment (K-12)` + frpm.
`Enrollment (Ages 5-17)` > 500;
```

This query sums the 'Enrollment (K-12)' and 'Enrollment (Ages 5-17)' columns and filters the schools with a total enrollment of over 500, returning the 'CDSCode' column.

Execution results

```
[['01100170109835'], ['01100170112607'], ['01100170124172']]
```

## I.2 MATH TASK

### Case Study-Math

System Prompt

Please reason step by step, and put your final answer within \\boxed{{}}.

User Prompt

Every morning Aya goes for a 9-kilometer-long walk and stops at a coffee shop afterwards. When she walks at a constant speed of $s$ kilometers per hour, the walk takes her 4 hours, including $t$ minutes spent in the coffee shop. When she walks $s + 2$ kilometers per hour, the walk takes her 2 hours and 24 minutes, including $t$ minutes spent in the coffee shop. Suppose Aya walks at $s + \frac{1}{2}$ kilometers per hour. Find the number of minutes the walk takes her, including the $t$ minutes spent in the coffee shop.

Response

To solve this problem, we need to first determine the value of $s$ and $t$. We know the following:

1. When Aya walks at $s$ km/h, the total time including the coffee shop visit is 4 hours. 2. When Aya walks at $s + 2$ km/h, the total time including the coffee shop visit is 2 hours and 24 minutes (which is 2.4 hours).

First, we can set up the equations for the walking times:

1. For the first scenario:

$$\frac{9}{s} + \frac{t}{60} = 4$$

2. For the second scenario:

$$\frac{9}{s + 2} + \frac{t}{60} = 2.4$$

We can solve these equations to find $s$ and $t$.

Let's subtract the second equation from the first to eliminate $t$:

$$\frac{9}{s} - \frac{9}{s + 2} = 1.6$$

This simplifies to:

$$\frac{9(s + 2) - 9s}{s(s + 2)} = 1.6$$

$$\frac{18}{s(s + 2)} = 1.6$$

$$18 = 1.6s(s + 2)$$

$$18 = 1.6s^2 + 3.2s$$
$$1.6s^2 + 3.2s - 18 = 0$$

We can solve this quadratic equation using the quadratic formula $s = \frac{-b \pm \sqrt{b^2 - 4ac}}{2a}$, where $a = 1.6$, $b = 3.2$, and $c = -18$:

$$s = \frac{-3.2 \pm \sqrt{3.2^2 - 4 \cdot 1.6 \cdot (-18)}}{2 \cdot 1.6}$$
$$s = \frac{-3.2 \pm \sqrt{10.24 + 115.2}}{3.2}$$
$$s = \frac{-3.2 \pm \sqrt{125.44}}{3.2}$$
$$s = \frac{-3.2 \pm 11.2}{3.2}$$

This gives us two possible solutions:

$$s = \frac{8}{3.2} = 2.5 \quad \text{(positive root)}$$

$$s = \frac{-14.4}{3.2} = -4.5 \quad \text{(negative root, not physically meaningful)}$$

So, $s = 2.5$ km/h. Now we can substitute $s$ back into one of the original equations to find $t$:

$$\frac{9}{2.5} + \frac{t}{60} = 4$$
$$3.6 + \frac{t}{60} = 4$$
$$\frac{t}{60} = 0.4$$
$$t = 24$$

So, $t = 24$ minutes. Now we need to find out how long the walk takes when Aya walks at $s + \frac{1}{2} = 2.5 + 0.5 = 3$ km/h. The total time including the coffee shop visit will be:

$$\frac{9}{3} + \frac{24}{60} = 3 + 0.4 = 3.4 \text{ hours}$$

Converting 3.4 hours to minutes, we get:

$$3.4 \times 60 = 204 \text{ minutes}$$

So the final answer is $\boxed{204}$.

## J PROOF OF THEOREM 1

The proof of Theorem 1 relies on bounding the total variation distance by various f-divergences. The following lemma, which is a direct consequence of the well-known Pinsker's inequality and its generalizations, establishes these relationships.

**Lemma 1.** *For any two probability distributions $P$ and $Q$ defined on the same measurable space, the squared total variation distance, $D_{TV}^2(P, Q)$, is bounded by the Kullback-Leibler (KL), $\alpha$-, and Jensen-Shannon (JS) divergences as follows:*

$$D_{TV}^2(P, Q) \leq C \cdot D_f(P\|Q)$$

*where $D_f$ is one of the specified divergences and $C$ is a positive constant that depends on the choice of $f$.*

*Proof.* The proof consists of recalling the standard Pinsker-type inequalities (Fedotov et al., 2003) for each specified f-divergence and arranging them into the desired form. We define the total variation distance as $D_{TV}(P, Q) = \frac{1}{2}\sum_x |P(x) - Q(x)|$.

- **Kullback-Leibler (KL) Divergence:** The classical Pinsker's inequality states that

$$D_{KL}(P\|Q) \geq 2D_{TV}^2(P, Q).$$

  Rearranging this directly yields the desired bound:

$$D_{TV}^2(P, Q) \leq \frac{1}{2}D_{KL}(P\|Q).$$

  Here, $C = 1/2$.

- **$\alpha$-Divergence:** The relationship depends on the value of $\alpha$.

  - For $\alpha > 1$, the $\alpha$-divergence is lower-bounded by the KL divergence. Consequently, the same constant applies:

$$D_\alpha(P\|Q) \geq D_{KL}(P\|Q) \geq 2D_{TV}^2(P, Q) \implies D_{TV}^2(P, Q) \leq \frac{1}{2}D_\alpha(P\|Q).$$

  - For $\alpha \in (0, 1)$, a similar quadratic bound holds, of the form $D_\alpha(P\|Q) \geq C_\alpha D_{TV}^2(P, Q)$ for some constant $C_\alpha > 0$. For instance, the Hellinger distance, which corresponds to $\alpha = 1/2$ (up to a factor), satisfies $H^2(P, Q) \geq 2D_{TV}^2(P, Q)$.

  In all cases, a suitable constant $C$ exists.

- **Jensen-Shannon (JS) Divergence:** For the JS divergence, there are two well-known lower bounds of this form.

  A commonly used quadratic inequality is:

$$D_{\text{JS}}(P\|Q) \geq \frac{1}{8}V(P, Q)^2$$

  The tightest possible bound overall is non-quadratic:

$$D_{\text{JS}}(P\|Q) \geq \ln(2) - H_2\left(\frac{1 + V(P, Q)}{2}\right)$$

  where $H_2(p) = -p\ln(p) - (1 - p)\ln(1 - p)$. This provides the sharpest possible function for $g(V)$.

  Thus, for each of the considered f-divergences, we have established the existence of a constant $C$ that satisfies the inequality. This concludes the proof.

$\square$

Throughout the following proof, we will repeatedly apply the lemma established in Schulman et al. (2015).

**Lemma 2** (Schulman et al. (2015)). *Given two policies $\pi$ and $\tilde{\pi}$, we have*

$$J(\tilde{\pi}) - J(\pi) = \mathbb{E}_{\tau \sim \tilde{\pi}}[\sum_{t=0}^{\infty} \gamma^t A_\pi(s_t, a_t)],$$

*where the expectation is over the trajectories $\tau := (s_0, a_0, s_1, a_1, ...)$ and the notion $\mathbb{E}$ indicates that $\tau$ are sampled from $\tilde{\pi}$ to generate $\tau$.*

Given that $\pi$, $\tilde{\pi}$ are $\alpha$-coupled policies, define $A^{\tilde{\pi}|\pi}(s) = \mathbb{E}_{\tilde{a} \sim \tilde{\pi}} A_\pi(s, \tilde{a})$ we have following result.

**Lemma 3** (Lemma 4 in Kang et al. (2018)). *let $(\pi, \tilde{\pi})$ be an $\alpha$ - coupled policy pair. Then*

$$|\mathbb{E}_{s_t \sim \tilde{\pi}}[A^{\tilde{\pi}|\pi}(s_t)]| \leq 2\alpha(1 - (1-\alpha)^t) \max_{s,a} |A_\pi(s, a)|$$

Now we are ready to prove our theorem 1.

*Proof.* Let's assume that the policy pairs $(\pi, \tilde{\pi})$ and $(\pi_{\text{pef}}, \tilde{\pi})$ are $\alpha$-coupled and $\beta$-coupled, respectively. This can be established if their total variation distances are bounded, i.e., $D_{TV}(\pi, \tilde{\pi}) \leq \alpha$ and $D_{TV}(\pi_{\text{pef}}, \tilde{\pi}) \leq \beta$. Our goal is to find a lower bound for the performance improvement $J(\tilde{\pi}) - L_\pi(\tilde{\pi})$. We proceed as follows:

$$J(\tilde{\pi}) - L_\pi(\tilde{\pi})$$

$$\overset{a}{=} \mathbb{E}_{s_t \sim \tilde{\pi}}[\sum_{t=0}^{\infty} \gamma^t A^{\tilde{\pi}|\pi}(s_t)] - \mathbb{E}_{s_t \sim \pi}[\sum_{t=0}^{\infty} \gamma^t A^{\tilde{\pi}|\pi}(s_t)]$$

$$= \sum_{t=0}^{\infty} \left( \mathbb{E}_{s_t \sim \tilde{\pi}}[\gamma^t A^{\tilde{\pi}|\pi}(s_t)] - \mathbb{E}_{s_t \sim \pi_{pef}}[\gamma^t A^{\pi_{pef}|\pi}(s_t)] + \mathbb{E}_{s_t \sim \pi_{pef}}[\gamma^t A^{\pi_{pef}|\pi}(s_t)] - \mathbb{E}_{s_t \sim \pi}(\gamma^t A^{\tilde{\pi}|\pi}(s_t)) \right)$$

$$\overset{b}{=} (\eta(\tilde{\pi}) - \eta(\pi)) - (\eta(\pi_{pef}) - \eta(\pi)) + \mathbb{E}_{s_t \sim \pi_{pef}}[\gamma^t A^{\pi_{pef}|\pi}(s_t)] - \mathbb{E}_{s_t \sim \pi}[\gamma^t A^{\tilde{\pi}|\pi}(s_t)]$$

$$= \eta(\tilde{\pi}) - \eta(\pi_{pef}) + \mathbb{E}_{s_t \sim \pi_{pef}}[\gamma^t A^{\pi_{pef}|\pi}(s_t)] - \mathbb{E}_{s_t \sim \pi}[\gamma^t A^{\tilde{\pi}|\pi}(s_t)]$$

$$\overset{c}{=} \sum_{t=0}^{\infty} \left( \mathbb{E}_{s_t \sim \tilde{\pi}}[\gamma^t A^{\tilde{\pi}|\pi_{pef}}(s_t)] + \mathbb{E}_{s_t \sim \pi_{pef}}[\gamma^t A^{\pi_{pef}|\pi}(s_t)] - \mathbb{E}_{s_t \sim \pi}[\gamma^t A^{\tilde{\pi}|\pi}(s_t)] \right)$$

$$\overset{d}{\geq} \sum_{t=0}^{\infty} \left( -2\gamma^t \beta(1 - (1-\beta)^t) \max_{(s,a)} |A_{\pi_{pef}}(s, a)| + \gamma^t \delta - 2\gamma^t \alpha(1 - (1-\alpha)^t) \max_{(s,a)} |A_\pi(s, a)| \right)$$

$$= -\frac{2\beta^2 \gamma \epsilon_p}{(1-\gamma)(1 - \gamma(1-\beta))} - \frac{2\alpha^2 \gamma \epsilon_\pi}{(1-\gamma)(1 - \gamma(1-\alpha))} + \frac{\delta}{1-\gamma}$$

$$\geq -\frac{2\gamma(\beta^2 \epsilon_p + \alpha^2 \epsilon_\pi)}{(1-\gamma)^2} + \frac{\delta}{1-\gamma}$$

(18)

where equality $a$, $b$, $c$ holds from Lemma 2; inequality $d$ applies Lemma 3.

The final result is obtained by using the fact that $(1 - \gamma(1-x)) > (1-\gamma)$ for $x \in (0, 1)$, which simplifies the denominator.

The theorem's full statement is established by using Lemma 1 to connect the KL-divergence bounds to the total variation distances ($D_{TV}$), which in turn provides the conditions for the policy pairs to be $\alpha$-coupled and $\beta$-coupled.

$\square$

