# OpenReview forum: "The Choice of Divergence: A Neglected Key to Mitigating Diversity Collapse in Reinforcement Learning with Verifiable Reward"
_ICLR.cc/2026/Conference — ICLR 2026 Poster_

### Official Review · Reviewer_HqbP · 2025-10-30

**Soundness:** 4
**Presentation:** 4
**Contribution:** 3
**Rating:** 8
**Confidence:** 5

**Summary:**

This paper addresses a critical paradox observed when fine-tuning large language models (LLMs) with Reinforcement Learning under Verifiable Rewards (RLVR): models often improve on single-attempt success (Pass@1) but see degraded multi-attempt performance (Pass@k) and suffer catastrophic forgetting of diverse skills. The authors identify that a key culprit is the choice of divergence in the RL objective. Prior approaches almost universally use a mode-seeking reverse KL divergence or even omit the divergence term, which provides no mechanism to retain broad knowledge. As a result, policies narrow toward a few high-probability solutions, sacrificing diversity.
To remedy this, the paper proposes Diversity-Preserving Hybrid RL (DPH-RL), a novel framework that uses mass-covering f-divergences (specifically forward-KL and Jensen–Shannon (JS) divergence) as a “rehearsal mechanism” to continually reference the initial policy. In effect, the divergence term itself becomes the solution: by penalizing divergence from the base model’s distribution, the agent is forced to maintain broad solution coverage and not forget previously learned behaviors. The approach is hybrid in that it selectively applies this divergence-based regularization, allowing unrestricted exploration on parts of the task where the base model wasn’t already proficient. Experiments on math problem solving and SQL query generation tasks demonstrate that DPH-RL not only improves in-domain Pass@1 accuracy, but crucially also boosts Pass@k (multiple attempts) performance above the base model’s level, while effectively preventing catastrophic forgetting on out-of-domain tasks. The method is also efficient: it computes the f-divergence via a generator function by sampling from the fixed initial policy, avoiding the need for an expensive separate reference policy network during training. Overall, this work highlights an overlooked axis in RL fine-tuning – the choice of divergence measure – as a powerful tool for building more general and diverse LLM-based problem solvers.

**Strengths:**

Important Problem & Insight: The paper tackles the important issue of diversity collapse in RL-based fine-tuning of LLMs, identifying an overlooked cause – the choice of KL divergence – and proposing it as a solution. This fresh insight (using forward-KL/JS to preserve knowledge) is highly original and addresses a real need.
Theoretical Rigor: The work is grounded in rigorous theory. The Enhanced Monotonic Improvement Theorem 1 generalizes TRPO’s guarantee to f-divergence regularization, showing a strictly better improvement bound with an added positive term from the reference policy. The assumptions and proof techniques are sound and build on well-established analysis, lending strong credibility to the approach.
Strong Empirical Results: The experiments are thorough and demonstrate clear benefits. DPH-RL consistently outperforms baseline methods (standard RL fine-tuning with reverse-KL, and variants without KL) by achieving higher Pass@k without sacrificing Pass@1. Notably, on an out-of-domain test (e.g. Spider SQL dataset), DPH-RL models maintain accuracy much closer to the base model than others, indicating greatly reduced forgetting. These results validate that the method fulfills its intended purpose of retaining diversity while improving performance.
Broad Applicability: The approach is relatively general. It’s compatible with different divergence choices (the authors demonstrate forward-KL and JS, and discuss extensions to a family of f-divergences) and can be plugged into existing RL algorithms (they test with GRPO, DAPO, etc.). This generality means the contribution could impact various subfields, from RLHF for LLMs to other generative modeling scenarios where mode-covering behavior is desired.
Clarity and Execution: The paper is well-written with clear motivation and analysis. Figures and tables strongly back the claims (for example, qualitative differences in solution diversity under different divergences are illustrated in figures). The method is described in sufficient detail, and the authors even address practical considerations (like how to compute the forward/JS divergence term efficiently by sampling the base model, which avoids needing an extra model and thus is training-efficient). This makes the work not only theoretically strong but also feasible and useful in practice.

**Weaknesses:**

While the paper presents a compelling and well-substantiated approach, a few limitations are worth noting:

Reliance on Base Policy Quality: The effectiveness of DPH-RL depends on the strength of the base model. If the base policy performs poorly on the target domain, there is little knowledge to rehearse, reducing the benefit of the divergence term.

Manual Partitioning: The hybrid framework splits training data into exploration and rehearsal sets using fixed thresholds on success rates. This manual setup introduces sensitivity to hyperparameters and may not adapt well across tasks without tuning.

Limited Divergence Exploration: Although the authors motivate the use of forward-KL and JS divergence, they do not evaluate a broader set of f-divergences or analyze tradeoffs across this family. It's unclear if the performance gains are tied to these specific choices or generalize more broadly.

Scope of Evaluation: The experiments are confined to two domains (math and SQL) and modest-sized models. Broader tasks and larger-scale settings would strengthen claims of generality.

Preservation of Undesirable Behavior: The method encourages retention of base model behavior, which may include suboptimal or biased responses. While not observed in current tasks, this risk should be acknowledged in future applications.

These issues do not undermine the core contributions but indicate valuable directions for further investigation and refinement.

**Questions:**

Adaptive vs. Fixed Data Partition: In your method, you partition the training data into D<sub>pef</sub> (where the base policy is nearly perfect) and D<sub>exp</sub> (exploration needed) using a fixed success-rate threshold. How sensitive are the results to this threshold choice, and did you consider a more dynamic criterion? For example, could the algorithm online-adjust the weight of the divergence term based on the model’s current performance on a state or task (rather than a hard split)? Such an adaptive approach might remove the need for manual threshold tuning; we’re curious if you experimented with or have thoughts on this extension.
Choice of Divergence Measure: You demonstrated significant gains with forward-KL and JS divergence. Did you try other f-divergences (e.g., intermediate alpha-divergences or Csiszár divergences) in practice, and if so, how did they compare? The theory suggests a family of options – is forward-KL essentially the strongest “mass-covering” extreme, or might a milder divergence also strike a good balance? Clarification on why forward-KL and JS were chosen (and whether one consistently outperforms the other in different conditions) would be helpful for practitioners looking to apply your framework.
Behavior with Minimal Base Knowledge: In scenarios where the base model has very low capability (e.g., near 0% success on D<sub>exp</sub> tasks), does DPH-RL effectively revert to a standard RL algorithm? Your analysis indicates that if the reference policy has no advantage (delta=0), the improvement bound reduces to the normal TRPO guarantee. Empirically, have you observed any downside to applying DPH-RL in such cases (perhaps the forward-KL term just stays inactive)? It would be reassuring to know that one at least loses nothing by using DPH-RL when the base model provides no guidance – confirming the method is robust in worst-case scenarios.
Generality to Other Settings: Can you comment on applying DPH-RL beyond the RLVR paradigm? For instance, in a standard RLHF setup (with a learned reward model and possibly no programmatic correctness signal), do you foresee any challenges in using a forward-KL/JS regularizer to preserve the base model distribution? Intuitively it should still help prevent policy collapse, but are there issues if the reward signal is less directly tied to correctness? Any insight or preliminary experiments in non-code, non-math tasks (e.g., dialogue) would be interesting to hear about.
Balancing Diversity and Quality: One might wonder if there are cases where preserving diversity could conflict with optimizing reward (e.g., if some of the base model’s diverse outputs are actually incorrect or suboptimal). Did you observe any instances where the forward-KL term kept the model generating suboptimal solutions for the sake of diversity? It seems your method focuses the preservation on high-performing reference behaviors (since D<sub>pef</sub> is constructed from successful attempts), which mitigates this. Still, any anecdotal observations on how DPH-RL balances keeping “good” diverse solutions versus discarding genuinely bad ones would provide intuition on the method’s nuances.

---

> ### Author Response · Authors · 2025-11-21
> **Response to Reviewer HqbP (Part 1)**
>
> We deeply appreciate your strong endorsement and insightful comments. Our response highlights five key aspects:
>
> **1. Sensitivity of Data Partitioning & Adaptive Strategy**
>
> To address this comprehensively, we have included detailed ablation studies in **Section 6.2** of the revised paper (please see the table below). The results demonstrate that our method exhibits strong robustness to threshold selection and consistently outperforms baseline models.
>
> * **Sensitivity Analysis:** Our experiments indicate negligible performance differences between the '7 of 8' and '8 of 8' settings, suggesting that the method is insensitive to the selection of high thresholds. While the '6 of 8' setting slightly boosts Greedy performance by focusing exploration on more challenging samples, further relaxing the threshold (e.g., to '5 of 8') degrades results due to the reduction in effective training data.
>
> | Model | Size of $D_{\text{exp}}$ | Size of $D_{\text{pef}}$ | Bird Greedy | Bird Pass@8 | Bird Pass@16 | Spider Greedy | Spider Pass@8 | Spider Pass@16 |
> | :---: | :---: | :---: | :---: | :---: | :---: | :---: | :---: | :---: |
> | GRPO | 8958 | 0 | 58.5 | 66.2 | 67.7 | 73.0 | 79.5 | 80.6 |
> | DAPO | 8958 | 0 | 60.0 | 67.2 | 69.0 | 71.1 | 75.3 | 76.7 |
> | 8 of 8 | 8266 | 692 | 61.4 | 71.6 | 73.5 | 75.4 | 86.3 | 87.2 |
> | 7 of 8 | 7429 | 1529 | 61.3 | 71.2 | 73.6 | 75.1 | 85.8 | 87.2 |
> | 6 of 8 | 6710 | 2248 | **62.8** | 70.5 | 72.4 | **76.0** | 82.7 | 84.1 |
> | 5 of 8 | 5783 | 3175 | 60.2 | 70.6 | 74.0 | 74.3 | 86.2 | 88.9 |
>
>
> * **Recommended Configuration & Mechanism Advantage:** Based on our experiments, we identify **DPH-JS (Threshold='8 of 8')** as the robust default configuration for practical applications. This leads to a critical insight: In standard GRPO, samples that are solved correctly in all 8 attempts typically yield **zero gradients** due to a lack of variance within the group, causing this high-quality data to be wasted. In contrast, DPH-RL effectively capitalizes on these "Mastered" samples—which are otherwise ignored by GRPO—by using them as anchors. Even though this subset constitutes only ~8% of the training data, by leveraging multiple generated trajectories per prompt to ensure diversity, DPH-RL consistently outperforms both GRPO and DAPO. This demonstrates that our method is **not only simple and parameter-free** but also statistically more efficient.
>
> * **Adaptive Strategy:** The **Online-Adaptive** approach you proposed is an excellent direction for extension. As you suggested, dynamically adjusting weights based on the model's current performance would not only eliminate manual threshold tuning but also smooth the boundary between $D_{pef}$ and $D_{exp}$ during training. We have prioritized this valuable concept as a core direction for our future work.
>
> **2. Behavior with Minimal Base Knowledge**
>
> **Theoretical Reversion & Robustness**: You are correct. In scenarios where the base model has minimal capability (resulting in an empty or near-empty D_pef), DPH-RL mathematically reverts to standard RL algorithms as the regularization term naturally deactivates. This ensures a "safety net," guaranteeing that our method is at least as effective as standard baselines in worst-case scenarios.
>
> **Empirical Advantage**: Furthermore, we argue that DPH-RL performs slightly better than standard RL even in these extreme cases. Standard RL often suffers from severe distribution shifts when valid signals are sparse. DPH-RL, however, acts as a stabilizer: once the model discovers even a rare correct trajectory, the Forward-KL/JS mechanism effectively "anchors" this precious signal, preventing it from being lost during the volatile exploration phase typical of low-resource settings.
>
> **3. Generality to Other Settings (RLHF)**
>
> **Transferability**: The principles of DPH-RL are fully transferable to the RLHF paradigm. While RLVR relies on programmatic correctness to construct D_pef, an RLHF setup can utilize Reward Model scores or human preference labels to select high-quality responses.
>
> **Mitigating Reward Hacking**: In RLHF, models often suffer from "Reward Hacking," collapsing into a single pattern to maximize scores. By applying DPH-RL, we can enforce a Forward-KL/JS constraint on high-reward samples. This ensures that the model optimizes for human preference while retaining the stylistic diversity and richness of the pre-trained distribution, preventing the policy from degrading into repetitive or unnatural outputs.

---

> > ### Author Response · Authors · 2025-11-21
> > **Response to Reviewer HqbP (Part 2)**
> >
> > **4. Choice of Divergence & Comparison with Alpha-Divergence**
> >
> > As you suggested, we did not limit our investigation to Forward-KL and JS divergence but further explored the broader family of f-divergences.
> >
> > * **Exploration of Alpha-Divergence:** We evaluated $\alpha \in \{0.2, 0.5, 0.8\}$ in our experiments. As shown in the table below, the results align with theoretical expectations: the performance of $\alpha$-divergence smoothly interpolates between Reverse-KL ($\alpha \to 0$, mode-seeking) and Forward-KL ($\alpha \to 1$, mass-covering). This confirms the rationale for focusing on Forward-KL and JS divergence, as they represent the critical points of "Strong Anchoring" and "Balanced State," respectively.
> >
> >     | Model | Bird Greedy | Bird Pass@8 | Bird Pass@16 | Spider Greedy | Spider Pass@8 | Spider Pass@16 |
> >     | :--- | :---: | :---: | :---: | :---: | :---: | :---: |
> >     | $\alpha=0.2$ | 60.1 | 67.0 | 69.1 | 75.4 | 78.8 | 80.1 |
> >     | $\alpha=0.5$ | 60.8 | 68.0 | 69.8 | 75.2 | 80.4 | 82.5 |
> >     | $\alpha=0.8$ | 60.5 | 69.2 | 70.4 | 75.8 | 81.7 | 83.6 |
> >
> > * **Unique Theoretical Advantages of JS Divergence:** Compared to Forward-KL, which purely pursues mass-covering, JS divergence exhibits unique **Symmetry** and **Boundedness**.
> >     * **Optimization Stability:** JS divergence is numerically strictly bounded ($0 \le D_{JS} \le \ln 2$). This results in a smoother optimization landscape compared to unbounded KL divergence, leading to more stable training.
> >     * **Balancing Mechanism:** Our goal is not for the model to converge to a single extreme, but to exhibit a **multi-modal distribution**: maintaining diversity on $D_{pef}$ while highlighting high-accuracy regions on $D_{exp}$. JS divergence effectively balances the "mode-seeking" nature of Reverse-KL and the "mass-covering" nature of Forward-KL, effectively preventing the policy from drifting toward extremes (i.e., preventing collapse).
> >
> > * **Empirical Comparison (DPH-JS vs. DPH-F):** We carefully compared the detailed performance curves of DPH-JS and DPH-F and found that DPH-JS outperforms or matches DPH-F on the vast majority of metrics.
> >
> > | Model | Greedy | Pass@8 | Pass@16 | pass@32 | pass@64 | pass@128 | pass@256 | OOD Avg |
> > | :--- | :---: | :---: | :---: | :---: | :---: | :---: | :---: | :---: |
> > | DPH-F | 60.4 | 70.1 | 71.6 | 75.2 | 77.8 | 79.6 | 81.5 | **60.98** |
> > | **DPH-JS** | **62.4** | **70.1** | **71.9** | **75.5** | **78.1** | **79.9** | **82.2** | 60.23 |
> >
> >
> > **5. Balancing Diversity and Quality**
> >
> > * Diversity as a Prerequisite for Optimization in GRPO: While diversity and reward maximization are often viewed as a trade-off, in the context of Group Relative Policy Optimization (GRPO), they are mutually reinforcing.
> >     * Avoiding Vanishing Gradients: GRPO relies on the relative advantage within a group of sampled responses. If diversity collapses and the model generates identical responses for a given query, the intra-group variance drops to zero. This leads to undefined advantages and vanishing gradients, effectively halting learning.
> >     * Conclusion: Therefore, by preserving diversity, DPH-RL ensures a continuous stream of effective training signals, which paradoxically aids in optimizing Pass@1, especially in later training stages.
> >
> > * No Retention of Suboptimal Solutions: We specifically designed the framework to avoid preserving incorrect behaviors.
> >     * Mechanism: The D_pef dataset is constructed strictly from verified correct trajectories.
> >     * Evidence: As shown in our Appendix experiment (Appendix B), the model learned 5 distinct styles of solving SQL problems. These solutions differed in approach (coverage) but yielded the same high accuracy. The divergence term encourages the model to master multiple valid reasoning paths ("flexible usage") rather than retaining incorrect or genuinely suboptimal solutions.

---

### Official Review · Reviewer_NVcc · 2025-10-31

**Soundness:** 4
**Presentation:** 3
**Contribution:** 4
**Rating:** 8
**Confidence:** 3

**Summary:**

This work studies the effect of divergence function choice on catastrophic forgetting, **pass@k** degradation, and diversity collapse in RL with verifiable rewards. The authors study the effects of reverse-KL on knowledge retention (described as performance preservation on tasks already solved before RLVR). Doing away with the mode-seeking reverse-KL, the authors propose a diversity-preserving framework that maintains task coverage and leads to both **pass@1** and **pass@k** improvements. The method has two stages: (1) a pre-sampling stage where the dataset is separated according to how solvable the problems are for the base model, (2) online training with two distinct loss functions—one without KL on the exploratory partition and another using mass-covering divergence functions on the mastered partition from (1). This is validated across Llama and Qwen models, showing great promise in mitigating the specialized performance vs. sampling diversity tradeoff.

**Strengths:**

-  **Line 86 to 88** : very good distinction of RL vs sequential generation with NNs. Important to disambiguate the problems between the two fields

- Clear introduction and preliminaries, easy to follow and well-motivated

- Addresses a core issue in the field

- No need to run inference on the reference model in the online training loop

**Weaknesses:**

- Relies on a good base model and lacks analysis of how balanced the partitions in the pre-sampling phase stage should be

- Figures axis are sometimes tough to read. Please use a standard plotting sofware with clear labels

- Unclear use of the term "stability"

- Trains only on two Qwen models which is insufficient given the claims of the paper

**Questions:**

1. What do the authors mean by in-domain and out-of-domain benchmarks at the end of the introduction?

2. Why is assumption 1 well-founded?

3. What happens if the partitions are not well-balanced? How possible is it and what are the contingencies?

4. In section 4.2, why is JS more stable? Is it due to it being more symmetric? Please clarify in either case.

---

> ### Author Response · Authors · 2025-11-21
> **Response to Reviewer NVcc (Part 1)**
>
> Dear Reviewer NVcc,
>
> We sincerely thank you for your recognition of our work and for your insightful feedback. We are encouraged by your positive evaluation of our approach. In this response, we address your concerns regarding the balance of data partitions with a comprehensive ablation study, clarify the theoretical stability of the JS divergence, and correct misconceptions regarding our model selection and domain definitions.
>
>  ## Response to W1 & Q3: Analysis of Partition Balance and Contingencies
>
> **W1 & Q3: Concerns regarding the balance of partitions in the pre-sampling phase and contingencies for unbalanced scenarios.**
>
> We appreciate the reviewer’s inquiry about the sensitivity of our method to partition balance. To address this, we conducted a comprehensive ablation study on the **Bird** and **Spider** benchmarks (see **Table R1**). We investigated the impact of varying the threshold for assigning data to the "performance-maintaining" dataset ($D_{\text{pef}}$) versus the "exploration" dataset ($D_{\text{exp}}$).
>
> *`8 of 8` indicates a threshold where only samples correctly solved in all 8 attempts are assigned to $D_{\text{pef}}$. `Full Data` implies using the entire dataset for both $D_{\text{pef}}$ and $D_{\text{exp}}$.*
>
> | Model | Size of $D_{\text{exp}}$ | Size of $D_{\text{pef}}$ | Bird Greedy | Bird Pass@8 | Bird Pass@16 | Spider Greedy | Spider Pass@8 | Spider Pass@16 |
> | :---: | :---: | :---: | :---: | :---: | :---: | :---: | :---: | :---: |
> | GRPO | 8958 | 0 | 58.5 | 66.2 | 67.7 | 73.0 | 79.5 | 80.6 |
> | DAPO | 8958 | 0 | 60.0 | 67.2 | 69.0 | 71.1 | 75.3 | 76.7 |
> | **8 of 8** | 8266 | **692** | 61.4 | 71.6 | 73.5 | 75.4 | 86.3 | 87.2 |
> | 7 of 8 | 7429 | 1529 | 61.3 | 71.2 | 73.6 | 75.1 | 85.8 | 87.2 |
> | 6 of 8 | 6710 | 2248 | **62.8** | 70.5 | 72.4 | **76.0** | 82.7 | 84.1 |
> | Full Data | 8958 | 8958 | 57.4 | **73.1** | **76.6** | 71.8 | **88.4** | **90.3** |
>
> **1. Impact of Imbalance and Robustness (Addressing W1):**
> Our method shows remarkable robustness even when partitions are extremely unbalanced.
> * In the `'8 of 8'` setting, $D_{\text{pef}}$ contains only **692 samples** (~7.7% of the original data). Despite this scarcity, it consistently outperforms baselines (GRPO and DAPO).
> * **Why it works:** The key objective is to preserve the *diversity* of the base model. Even if the number of unique questions in $D_{\text{pef}}$ is small, we maximize diversity by **retaining all sampled trajectories** (e.g., all 8 generations) for these questions. This ensures that the distribution is well-anchored even with fewer prompts.
>
> **2. Contingencies for Extreme Imbalance (Addressing Q3):**
> The reviewer asks about contingencies if partitions are not well-balanced (e.g., $D_{\text{pef}}$ is extremely small).
> * **Contingency Plan:** As demonstrated in the `'8 of 8'` experiment, our contingency for low data volume is **trajectory expansion**. By utilizing *all* generated trajectories for the few available high-quality prompts (instead of filtering for a single best one), we provide sufficient distributional coverage to regularize the model.
> * **Superiority over Baselines:** Crucially, samples that are correctly solved 8 out of 8 times typically yield **zero gradients** in standard GRPO due to a lack of variance (baseline $\approx$ reward). Our method effectively capitalizes on these "mastered" samples—which are otherwise wasted—offering a statistically unbiased solution that stabilizes training even when such samples are rare.
>
> **3. Failure Mode of "Perfect Balance" (Full Data):**
> Conversely, enforcing a "balanced" full overlap (`Full Data`) actually degrades greedy performance (e.g., Bird Greedy drops to 57.4). This failure stems from:
> 1.  **Statistical Bias:** Including rare successes (e.g., 1/8 correct) in the reference distribution preserves noise rather than high-quality capabilities.
> 2.  **Trivial Optimization:** The model minimizes loss by simply mimicking a single correct answer without genuine learning, hurting generalization.
>
> **Conclusion:** DPH-RL demonstrates robust capabilities across various distributions of $D_{\text{pef}}$. Despite slight performance variations among settings, all consistently outperform GRPO and DAPO. We recommend starting with the **'8 of 8'** setting. When encountering imbalance, we leverage multiple generated trajectories per prompt to ensure diversity; if the data remains insufficient, the threshold for partitioning $D_{\text{pef}}$ and $D_{\text{exp}}$ can be adjusted to construct a larger reference dataset.

---

> > ### Author Response · Authors · 2025-11-21
> > **Response to Reviewer NVcc (Part 2)**
> >
> > **W2: Figures axis are sometimes tough to read. Please use a standard plotting sofware with clear labels**
> >
> > We thank the reviewer for the feedback. We have updated **Figure 2 and Figure 3** in the revised version to ensure they are clearly legible.
> >
> > **W3 & Q1: Clarification on the Stability of JS Divergence**
> >
> > The term "stable" here implies two distinct meanings:
> >
> > 1.  **The inherent stability of JS divergence itself:**
> >     The calculation of JS divergence involves comparing the new policy $\pi_{\theta}$ and the reference policy $\pi_{ref}$ respectively against their average mixture distribution $M = \frac{\pi_{\theta} + \pi_{ref}}{2}$. Since the mixture distribution $M$ contains probability mass from both distributions, the denominator will never be zero; consequently, the divergence term will never "explode." Numerically, it is strictly bounded. This results in a smoother and more stable optimization landscape during the training process.
> >
> > 2.  **Symmetry and Balance:**
> >     JS divergence is symmetric. This means it balances the "mode-seeking" characteristic of Reverse-KL and the "mass-covering" characteristic of Forward-KL, thereby effectively preventing the model from drifting towards extremes (i.e., preventing policy collapse). regarding the training results, our goal is not for the model to converge to a single extreme; rather, we hope the model exhibits a multi-modal form, maintaining diversity on the original $D_{pef}$ distribution while highlighting high-accuracy regions on $D_{exp}$. JS divergence effectively balances mass-covering and mode-seeking.
> >
> > **W4: Trains only on two Qwen models which is insufficient given the claims of the paper**
> >
> > We respectfully clarify that our experiments are not limited to Qwen models. As shown in **Table 1**, we utilized **both Llama-3.1 and Qwen** models. In fact, the majority of our experimental analysis was conducted on Llama-3.1.
> >
> > **Q1: What do the authors mean by in-domain and out-of-domain benchmarks at the end of the introduction?**
> >
> > **In-domain** refers to test data that follows the same distribution as the RL training dataset, whereas **out-of-domain** refers to data from distinct fields or tasks. In our SQL experiments, we exclusively utilized the BIRD dataset for training. Consequently, the BIRD test set constitutes the **in-domain** benchmark, while mathematical datasets—which represent tasks completely distinct from SQL generation—are considered **out-of-domain**.
> >
> > **Q2: Why is assumption 1 well-founded?**
> >
> > Regarding the reasonableness of Assumption 1 in Section 4.3, it stems from the fact that our framework dynamically adjusts the definition of the conditional reference policy $\pi_{pef}$ based on the data type ($D_{pef}$ and $D_{exp}$), ensuring the assumption holds in both scenarios. The core requirement of this assumption is that for any state $s$, the expected advantage of the reference policy's actions relative to the current policy $\pi$ has a non-negative lower bound $\delta \ge 0$.
> >
> > Specifically, on the near-perfect dataset $D_{pef}$, $\pi_{pef}$ is defined as the policy induced by the stored correct trajectories. Since $\pi_{pef}$ represents established expert behavior, its actions are expected to be superior to those of the current policy $\pi$ (which is still learning), naturally yielding a positive advantage. Therefore, it is reasonable to assume $\delta > 0$ in this context, as it reflects the process of learning from expert demonstrations.
> >
> > On the exploration dataset $D_{exp}$, we define the reference policy $\pi_{pef}$ as the current policy $\pi$ itself. When the reference policy is identical to the current policy, the expected advantage of the policy with respect to itself is inherently zero. Consequently, the inequality trivially holds here with $\delta = 0$, which corresponds to the model performing unrestricted exploration without regularization constraints.
> >
> > In summary, Assumption 1 acts as a unified mathematical expression that cleverly encompasses both "learning from experts" (Advantage > 0) and "self-exploration" (Advantage = 0), making it fully valid within the DPH-RL framework.

---

> > > ### Comment · Reviewer_NVcc · 2025-11-26
> > >
> > > Thank you for your answer. Apologies on the typed oversight regarding the models used, I meant to type that more models in addition to Qwen **and** Llama would be better, but the value of  your contribution still stands.

---

> > > > ### Author Response · Authors · 2025-11-27
> > > > **Reply to Reviewer NVcc**
> > > >
> > > > Thank you for your reply, and thank you again for your recognition of our work!

---

### Official Review · Reviewer_8NDo · 2025-11-03

**Soundness:** 2
**Presentation:** 3
**Contribution:** 2
**Rating:** 4
**Confidence:** 4

**Summary:**

This paper re-exames the KL penalty term in RLVR with LLMs. Most approaches adopt the reverse-KL, which may degrade LLMs’ Pass@K performance. To mitigate this problem, this paper proposes to use forward-KL and consider a more general divergence family, f-divergences.

**Strengths:**

1. This paper studies an important and interesting problem in RLVR that how the KL divergence affects RL training.

**Weaknesses:**

1. The motivation for forward-KL is not strong enough.

    The failure of reverse-KL does not point to forward-KL or JS divergence, especially when authors did not analyze the failure reason for reverse-KL.

2. The lack of discussion with replay buffer.

    The implementation of forward-KL requires collecting trajectories from the base model, which can be consider as a special rehearsal mechanism. Replay buffers and their application are widely studied in both of continual learning and RL. Some discussions of replay buffer may clarify this paper’s position.

3. The necessity of splitting dataset into “near-perfect” or “exploration”.

    > For queries where π_ref performs well, aggressive reward maximization is unnecessary and risks degrading performance.

    If this claim is true, then any RL post-training should not include “easy” questions. However, LLMs tend to forget prior learned knowledge. Adding easy questions can help to remind LLMs, aligning author’s intention of rehearsal-stylish forward-KL.

4. The lack of in-depth analysis of forward-KL.

    Though forward-KL seems to achieve empirical success, its mechanism is still under unexplored. Although authors provide the Monotonic Improvement of the method, it is hard to connect it with the improvement. Moreover, the discussion in section 6 only shows a ratio between training, which does not contain too much information.

5. Authors should set a pointer to explicitly link the definition of alpha-divergence.

**Questions:**

See the above weakness.

Additionally:
1. What does “greedy” mean in the experiment section? Pass@1 or greedy decoding?
2. What is $\pi_{\text{per}}$ in section 4.3?
3. What does the alpha in Table 2 mean? alpha-divergence?
4. If our goal is to improve pass@K performance and we know that the base model has the highest pass@K performance, then what is the point of RL post-training?
5. Do authors have any idea why RL training hurts pass@K performance?

---

> ### Author Response · Authors · 2025-11-21
> **Response to Reviewer 8NDo (Part 1)**
>
> Dear Reviewer 8NDo,
>
> We sincerely thank the reviewer for the insightful and constructive comments. We have carefully considered all points, particularly regarding the motivation for Forward-KL, the necessity of the dataset splitting strategy, and the analysis of the diversity preservation mechanism. Below, we address each concern with detailed clarifications and new experimental evidence.
>
> ---
>
> ## W1 & W4: Motivation, Failure Analysis, and Mechanism of Forward-KL
>
> **Response:**
>
> We respectfully argue that our paper provides a solid theoretical foundation and direct empirical evidence linking the failure of **both Reverse-KL and "No-KL"** approaches specifically to the motivation for Forward-KL. Furthermore, **we clarify a crucial misunderstanding regarding Section 6: it is not an analysis of training data ratios, but a fundamental decomposition of the learning mechanism on test data (Retention vs. Exploration).**
>
>  ### 1. Mechanism Analysis of Forward-KL (Clarification on Section 6)
>
> We conducted a **mechanism decomposition of performance sources** to explain *why* Forward-KL succeeds where others fail.
>
> * **Keep vs. Explore:** We decomposed the Pass@k score into two distinct components:
>     * **Keep Rate (Knowledge Consolidation):** The percentage of problems the Base Model could already solve that the RL-trained model *still* solves correctly.
>     * **Additional Exploration Rate (New Knowledge):** The percentage of problems the Base Model failed that the RL-trained model now solves.
> * **Mechanism of Success:**
>     * **Figure 3** reveals the critical flaw in baselines: Reverse-KL and No-KL methods suffer a massive drop in **Keep Rate** (e.g., retaining only ~85% of original capabilities on OOD tasks). This confirms they suffer from **catastrophic forgetting**.
>     * **DPH (Forward-KL/JS)** maintains a significantly higher Keep Rate while achieving comparable or superior Exploration, as it continues to receive more valid reward signals during the later stages of training.
> * **Conclusion:** This empirically proves the specific mechanism of Forward-KL: it improves overall performance not just by learning new paths, but by functioning as a **"rehearsal mechanism"** that prevents the forgetting of diverse solutions inherent in the Base Model.
>
>  ### 2. Motivation for Forward-KL and Failure Analysis of Baselines
>
> Based on the mechanism identified above, the motivation for Forward-KL is established through the specific failures of the alternatives:
>
> * **Theoretical Foundation (Mode-Seeking vs. Mass-Covering):**
>     * **Reverse-KL Failure:** As discussed in the Introduction, standard Reverse-KL is inherently **mode-seeking**. It actively penalizes the model for maintaining probability mass on valid solutions that are not the single highest-probability peak.
>     * **Forward-KL Solution:** Forward-KL is **mass-covering**. It penalizes the policy for *failing to cover* the high-probability modes of the reference policy, thereby theoretically enforcing the "Keep Rate" we identified as crucial in Section 6.
>
> * **Empirical Evidence of Mode Collapse (Figure 4):**
>     We explicitly analyzed the failure mode of Reverse-KL through a targeted diversity experiment detailed in Appendix B and Figure 4, which is also highlighted in our introduction. We constructed a base model capable of generating five distinct solution styles for SQL problems and subjected it to RL training. To evaluate diversity, we sampled 32 responses for each query and counted the frequency of each distinct style based on output prefixes, categorizing unrecognized patterns as "Others". As illustrated in Figure 4, the Base Model exhibits high diversity, producing multiple solution styles in 81% of cases. However, training with GRPO (Reverse-KL) leads to severe mode collapse, with the model degenerating to a single style in 94% of cases. In sharp contrast, DPH-F (Forward-KL) successfully preserves the original diversity, showing single-style collapse in only 3% of cases. This visualization provides concrete empirical evidence that the failure of Reverse-KL stems from mode collapse, whereas Forward-KL effectively mitigates this issue by enforcing distributional coverage.
>
> * **Failure of "No-KL" Methods:**
>     * The reviewer suggests the failure of Reverse-KL does not imply Forward-KL. However, we demonstrated that simply removing the KL term (i.e., "No-KL" methods like GRPO w/o KL or DAPO) is also fatal.
>     * **Table 1** shows that methods lacking a divergence constraint experience severe Pass@k collapse, often falling below the Base Model (e.g., Base Model Pass@8 is 68.8%, while GRPO/DAPO drop to 66.2%/67.2%), whereas on Pass@8, the Reverse-KL and DPH series methods all outperform the Base Model.
>     * **Figure 3** confirms this is due to the model drifting away from the diverse knowledge base (low Keep Rate). Thus, a constraint is necessary, but it must be the *correct* constraint.

---

> > ### Author Response · Authors · 2025-11-21
> > **Response to Reviewer 8NDo (Part 2)**
> >
> > ## W2: Discussion on Replay Buffer
> > > *Reviewer Comment: The lack of discussion with replay buffer... implementation of forward-KL ... can be considered as a special rehearsal mechanism.*
> >
> > We thank the reviewer for this insightful comment. We fully agree that the implementation of Forward-KL (DPH-F) serves as a rehearsal mechanism, and we appreciate you pointing out the connection to Replay Buffers in Continual Learning (CL).
> >
> > **Connection to Rehearsal Mechanism:**
> >
> > You are right. Minimizing the Forward-KL divergence ($E_{x \sim \pi_{ref}} [-\log \pi_{\theta}(x)]$) is mathematically equivalent to maximizing the likelihood of trajectories sampled from the reference policy. This acts effectively as an Experience Replay (ER) buffer containing high-quality solutions ("perfect" trajectories), preventing the model from catastrophically forgetting previously mastered skills while optimizing the reward on new data.
> >
> > **While** Forward-KL acts as a direct rehearsal, our DPH-JS (Jensen-Shannon) introduces a more nuanced mechanism. Since JS divergence is symmetric and involves the mixture distribution $M = (\pi_{ref} + \pi_{\theta})/2$, it requires sampling from both the collected trajectories (the buffer) and the current policy's self-generated trajectories.
> > This makes DPH-JS a form of "Regularized Rehearsal":
> > It anchors the model to the reference distribution (like a Replay Buffer).
> > Simultaneously, it prevents the model from drifting too far in its own exploration, but without forcing it to purely "memorize" the buffer as strictly as Forward-KL/SFT might.
> >
> > Revision Action:
> >
> > Per your suggestion, we have added a discussion regarding the differences between DPH-RL and replay buffer methods in Related Work (p2, line 134-137) and Appendix D (p16-p17).

---

> ### Author Response · Authors · 2025-11-21
> **Response to Reviewer 8NDo (Part 3)**
>
> ## W3: Necessity of Splitting Dataset ($ D_{pef} $ vs $ D_{exp} $)
> > *Reviewer Comment: The necessity of splitting dataset... aggressive reward maximization is unnecessary... Adding easy questions can help to remind LLMs...*
>
> **Response:**
>
> We appreciate this insightful comment. However, we wish to clarify that our partitioning strategy is not aimed merely at maximizing "correctness," but specifically at **preserving the diversity of the reference model on correct solutions**.
>
> We address your concern with three key arguments supported by new experimental evidence:
>
> | Model | Size of $D_{exp}$ | Size of $D_{pef}$ | Bird Greedy | Bird Pass@8 | Spider Greedy | Spider Pass@8 |
> | :--- | :---: | :---: | :---: | :---: | :---: | :---: |
> | GRPO | 8958 | 0 | 58.5 | 66.2 | 73.0 | 79.5 |
> | DAPO | 8958 | 0 | 60.0 | 67.2 | 71.1 | 75.3 |
> | **Ours (Split)** | 6710 | 2248 | **62.8** | 70.5 | **76.0** | 82.7 |
> | Full Data | 8958 | 8958 | 57.4 | 73.1 | 71.8 | 88.4 |
>
> **1. Inclusion of Easy Questions Does Not Guarantee Diversity**
>
> Merely including easy questions in the $\text{RL}$ policy update data ($D_{exp}$) will not prevent diversity collapse. Standard methods like GRPO and DAPO utilize the complete dataset (comprising both easy and hard questions) for reward maximization. Despite "seeing" all easy questions, these models still suffer from severe **diversity collapse** (as evidenced by the sharp drop in Pass@k). Optimizing solely for reward on easy questions encourages the model to converge to a deterministic, high-reward mode rather than preserving the original diverse distribution.
>
> **2. Statistical Justification: Unbiased vs. Biased Estimation**
> The rationale for partitioning lies in the statistical nature of sampling "correct" responses to form the reference.
> * **On Near-Perfect Data ($D_{pef}$):** The base model solves these problems with high probability (e.g., 8/8). Here, the empirical distribution of $\pi_{ref}$ closely approximates the true diverse distribution of correct solutions ($ P^* $):
>     $\pi_{ref}(y|x) \approx P^*(y|x)$
> * **On Hard Data:** For problems with low success rates, filtering for correctness introduces massive statistical bias.
>
> To validate this, we conducted a **new experiment** applying the DPH-JS objective to the **Full Data** (treating all samples as $D_{pef}$ for the divergence constraint).
>
> * **Failure of Full Data:** When using the entire dataset for the divergence constraint (treating all data as $D_{pef}$), the **Greedy (Pass@1)** performance drops significantly to **57.4%**. This is even lower than the GRPO baseline (**58.5%**) and far below our optimal split strategy (**62.8%**).
>
> * **Reasons for Splitting:**
>     1.  **Statistical Bias & Noise:** For "hard" problems (e.g., solved only 1/8 times by the base model), forcing the model to treat that single stochastic success as a "Reference" for Forward-KL introduces huge statistical bias.
>     2.  **Trivial Mimicry:** If simple questions are mixed into the exploration set without proper constraints, or if hard questions are treated as reference, the model effectively maximizes reward and minimizes KL by simply overfitting to the single known solution. This optimizes the loss metric without learning generalizable reasoning.
>
>
> ## W5: Pointer to Alpha-divergence
>
> We thank the reviewer for pointing this out. The definition of $\alpha$-divergence is located in **Table 5 of Appendix C**. We added an explicit pointer in the main text to link to this definition in the revised version.

---

> > ### Author Response · Authors · 2025-11-21
> > **Response to Reviewer 8NDo (Part 4)**
> >
> > **Q1: What does “greedy” mean in the experiment section?**
> >
> > **Response:** "Greedy" refers to the use of greedy decoding (temperature=0). While Pass@k typically employs a temperature > 0 to ensure diversity, when $k=1$, greedy decoding often reflects the model's single-attempt capability better than sampled Pass@k.
> >
> > **Q2: What is $\pi_{per}$ in section 4.3?**
> >
> > **Response:** **Q2: What is $\pi_{per}$ in section 4.3?**
> >
> > **Response:**
> >
> > $\pi_{per}$ represents the ideal target distribution used in our Forward-KL objective. As detailed in Section 4.3, it is defined conditionally based on the data split:
> >
> > * **On the near-perfect dataset ($D_{pef}$):** $\pi_{per}$ is the policy induced by the **stored correct trajectories** from the base model. This acts as the reference for Forward-KL to enforce "mass-covering" behavior and prevent the forgetting of known solutions.
> > * **On the exploration dataset ($D_{exp}$):** $\pi_{per}$ is defined as the **current policy $\pi$**.
> >
> > **Q3: What does the alpha in Table 2 mean?**
> >
> > **Response:** Yes, it refers to the $\alpha$ parameter in $\alpha$-divergence. We **added** a clarification in Table 2.
> >
> > **Q4: If the Base Model has the highest Pass@k, what is the point of RL post-training?**
> >
> > **Response:**
> > The primary goal of RL post-training is to make the model usable by drastically boosting Pass@1.
> > * **Limitation of Base Model:** While the Base Model has high diversity (high Pass@k), its Pass@1 is often unacceptably low (e.g., **42.4%** on SQL in Table 1). This makes it practically unusable for single-turn applications.
> > * **Role & Cost of RL:** Traditional RL (like GRPO) boosts Pass@1 to a usable level (e.g., ~60%), but at the cost of a severe drop in Pass@k (diversity).
> > * **Our Contribution:** DPH-RL boosts Pass@1 to the same high standard as GRPO, while significantly mitigating the diversity collapse (Pass@k degradation) relative to the Base Model. We transform a 'diverse but unreliable' model into one that is 'reliable (high Pass@1) and robust,' preserving the vast majority of the Base Model's original diversity.
> >
> > **Q5: Do authors have any idea why RL training hurts Pass@k performance?**
> >
> > **Response:**
> > In the context of RL for LLMs, [4] conceptualize the base model as a tree structure. While the initial model is capable of navigating various branches, policy optimization leads the model to discover that focusing on specific paths rapidly improves accuracy. This phenomenon is described as the "sharpening" effect, wherein the model over-exploits a small subset of high-reward paths to boost Pass@1. Similarly, [5] is noted that to maximize rewards, RL algorithms rapidly sacrifice policy entropy. This causes the policy to become overly deterministic, thereby diminishing its exploration capabilities. Although the authors observed that global entropy calculations might be imprecise due to implementation discrepancies between vLLM and FSDP during sampling—or because entropy shifts are more dependent on specific pivotal tokens—we observed [6] qualitatively that the model's outputs became increasingly homogenous during training.
> >
> > While it is typically understood that improving Pass@1 requires sacrificing some degree of Pass@k, the philosophy behind DPH-RL is to prevent over-optimization on Pass@1, aiming to maintain high performance on both Pass@1 and Pass@k simultaneously.
> >
> > [1] Ouyang L, Wu J, Jiang X, et al. Training language models to follow instructions with human feedback[J]. Advances in neural information processing systems, 2022, 35: 27730-27744.
> >
> > [2] Zhang H, Fu J, Zhang J, et al. Rlep: Reinforcement learning with experience replay for llm reasoning[J]. arXiv preprint arXiv:2507.07451, 2025.
> >
> > [3] Li Z, Xu T, Zhang Y, et al. ReMax: a simple, effective, and efficient reinforcement learning method for aligning large language models[C]//Proceedings of the 41st International Conference on Machine Learning. 2024: 29128-29163.
> >
> > [4] Yue Y, Chen Z, Lu R, et al. Does reinforcement learning really incentivize reasoning capacity in llms beyond the base model?[J]. arXiv preprint arXiv:2504.13837, 2025.
> >
> > [5] Cui G, Zhang Y, Chen J, et al. The entropy mechanism of reinforcement learning for reasoning language models[J]. arXiv preprint arXiv:2505.22617, 2025.
> >
> > [6] Wang S, Yu L, Gao C, et al. Beyond the 80/20 rule: High-entropy minority tokens drive effective reinforcement learning for llm reasoning[J]. arXiv preprint arXiv:2506.01939, 2025.

---

> ### Author Response · Authors · 2025-11-27
> **Official Comment by Authors**
>
> We are very grateful for your thoughtful and detailed feedback on our manuscript. We have prepared a detailed response.
>
> As the discussion period is drawing to a close, please do not hesitate to let us know if there are any further questions or areas for clarification. We are ready and willing to discuss any remaining concerns. We thank you once more for your invaluable contribution to our work.

---

### Official Review · Reviewer_3S5Q · 2025-11-10

**Soundness:** 2
**Presentation:** 2
**Contribution:** 2
**Rating:** 4
**Confidence:** 3

**Summary:**

This paper proposes a divergence-based policy optimization framework that replaces the conventional reverse KL term with forward KL or Jensen–Shannon (JS) divergence. The authors argue that forward KL can mitigate mode-seeking behavior and stabilize updates. Experiments on two representative models demonstrate improved training stability and comparable or better performance.

**Strengths:**

* The proposed method is conceptually simple and easy to implement.
* Theoretical motivation is clear: reinterpreting divergence choice (reverse vs. forward KL) in policy optimization.
* Experimental results on two models provide empirical support for the claimed benefits.

**Weaknesses:**

* The main idea—switching from reverse KL to forward KL or JS divergence—has been extensively discussed in prior work [1]. The paper should (a) more directly situate DPH-RL relative to that prior theory, (b) clarify what is new beyond applying these known divergence properties to RLVR for LLMs, and (c) compare empirically to algorithms motivated by those analyses.

[1] Chan A, Silva H, Lim S, et al. Greedification operators for policy optimization: Investigating forward and reverse kl divergences[J]. Journal of Machine Learning Research, 2022, 23(253): 1-79.

**Questions:**

* How sensitive are the results to the choice of temperature or scaling in the KL term?
* Did you tune the KL coefficients separately for reverse and forward cases to ensure fairness?
* Can the method be applied to off-policy or batch RLHF settings, where distribution mismatch is more severe?

---

> ### Author Response · Authors · 2025-11-21
> **Response to Reviewer 3S5Q (Part 1)**
>
> Dear Reviewer 3S5Q,
>
> We thank the reviewer for pointing out the relevant work by Chan et al. [1]. We agree that [1] provides a foundational theoretical analysis of divergences in policy optimization. However, DPH-RL differs fundamentally from the framework in [1] in terms of optimization objective, algorithmic derivation, and implementation paradigm.
>
>  ## 1. Difference in Objective and Role of Divergence (Addressing a & b)
>
> The core distinction lies in the mathematical formulation and the specific role the divergence plays in the objective function:
>
> **Chan et al. [1] (Divergence as a Greedification Operator):**
> The framework in [1] aligns with the standard RL objective of maximizing expected reward:
>
> $ \max_{\pi} \mathbb{E}_{\pi}[r] $
>
> To achieve this, they use KL divergence as a "Greedification Operator" to project the policy towards a target Boltzmann distribution induced by Q-values ($\mathcal{B}^{\tau} Q$). The update is formulated as minimizing either the Reverse or Forward KL divergence:
>
> $ \pi_{new} = \arg\min_{\pi} D_{KL}(\pi \parallel \mathcal{B}^{\tau} Q) \quad \text{(Reverse)} \quad \text{or} \quad \arg\min_{\pi} D_{KL}(\mathcal{B}^{\tau} Q \parallel \pi) \quad \text{(Forward)}$
>
> Here, the divergence is the driving force that pulls the policy towards high-reward regions.
>
> **DPH-RL (Divergence as a Regularization Constraint):**
> In contrast, we explicitly modify the objective to include a retention constraint. We formulate the problem as:
>
> $max_{\pi} E_{\pi}[r] - \eta D_{f}(\pi || \pi_{ref})$
>
> Here, the divergence acts as an anchor rather than a target. We leverage the mass-covering property of Forward KL (or other $f$-divergence) to constrain the policy to retain the solution space of the reference model ($\pi_{ref}$). This regularization term is critical for preventing the diversity collapse that typically occurs when solely optimizing for $\max \mathbb{E}[r]$ or using aggressive greedification.
>
> ## 2. Difference in Algorithmic Derivation (Addressing b)
>
> These differing objectives lead to distinct update rules:
>
> **Chan et al. [1]:** As summarized in their Table 1, the choice of divergence dictates the algorithm:
> * **Hard Reverse KL** corresponds to the standard REINFORCE algorithm (Policy Gradient).
> * **Hard Forward KL** corresponds to Rejection Sampling, where the policy is updated to strictly mimic the highest-reward samples.
>
> **DPH-RL:** In contrast, our update rule is fundamentally different. As defined in Equation (7) of our paper, we combine a GRPO/DAPO loss with an $f$-divergence regularization term:
>
> $L_{DPH-RL}(\theta) = L_{exp}(\theta) + \eta L_{pef}(\theta)$
>
> This differs significantly from Rejection Sampling. Instead of discarding samples to fit a target distribution, DPH-RL optimizes the policy via gradient ascent on advantages ($L_{exp}$) while simultaneously using the divergence term ($L_{pef}$) as a soft anchor to preserve the reference model's capabilities.
>
>  ## 3. Empirical Comparison to Algorithms Motivated by Prior Analysis (Addressing c)
>
> To empirically validate this distinction, we followed the reviewer’s suggestion to compare DPH-RL against the algorithms motivated by [1]. We explicitly conducted an additional experiment implementing the "Hard FKL" algorithm as defined in [1], which treats the action with the maximum Q-value as the target label (conceptually similar to Rejection Sampling Fine-Tuning).
> We compared this against GRPO (equivalent to Hard Reverse KL in [1]) and our proposed DPH-F.
>
> | Model | Bird Greedy | Bird Pass@8 | Bird Pass@16 | Spider Greedy | Spider Pass@8 | Spider Pass@16 |
> | :--- | :---: | :---: | :---: | :---: | :---: | :---: |
> | Hard FKL | 52.2 | 55.6 | 58.7 | 67.9 | 72.6 | 73.1 |
> | GRPO (Hard RKL) | 58.5 | 66.2 | 67.7 | 73.0 | 79.5 | 80.6 |
> | DPH-F (Ours) | 60.4 | 70.1 | 71.6 | 77.4 | 84.5 | 85.7 |
>
> **Results and Analysis:**
> The results demonstrate that the implementation paradigm matters significantly:
>
> * **Performance Gap:** Hard FKL performs significantly worse than DPH-F. On the in-domain Bird dataset, Hard FKL achieves only 52.2% Greedy accuracy, compared to 60.4% for DPH-F.
> * **Diversity Collapse:** The decline is even more pronounced in diversity metrics. Hard FKL's Pass@16 is 58.7%, while DPH-F maintains 71.6%.
>
> These results confirm that using Forward KL as a greedification operator (Hard FKL) leads to rapid diversity collapse by forcing the model to mimic a single solution path. In contrast, DPH-F successfully uses Forward KL as a retention mechanism on the $\mathcal{D}_{pef}$ dataset, preserving the reference model's solution space and achieving superior performance.

---

> > ### Author Response · Authors · 2025-11-21
> > **Response to Reviewer 3S5Q (Part 2)**
> >
> > **Q1: How sensitive are the results to the choice of temperature or scaling in the KL term?**
> >
> > **Q2: Did you tune the KL coefficients separately for reverse and forward cases to ensure fairness?**
> >
> > Response to Q1&Q2:
> >
> > In our experiments, the "temperature or scaling" in the KL term and the "KL coefficient" refer to the same parameter, specifically the hyperparameter $\eta$ (Eq. 4 and Eq. 7), which controls the weight of the $f$-divergence regularization term. We explicitly analyzed the sensitivity to $\eta$ in Section 5.1 (Ablation Study, p7) and Figure 2(b). To address your concern more clearly, we tested two additional sets of parameters.
> >
> > | Model | Bird Greedy | Bird Pass@8 | Bird Pass@16 | Spider Greedy | Spider Pass@8 | Spider Pass@16 |
> > | :--- | :---: | :---: | :---: | :---: | :---: | :---: |
> > | *DPH-JS (Generator)* | | | | | | |
> > | -$\eta$=0.01 | 59.8 | 66.7 | 68.1 | 76.4 | 79.1 | 80.8 |
> > | -$\eta$=0.05 | 61.3 | 69.5 | 70.9 | 75.3 | 82.7 | 83.8 |
> > | -$\eta$=0.1 | 62.2 | 70.3 | 71.8 | 75.6 | 83.6 | 83.9 |
> > | -$\eta$=0.2 | 62.8 | 70.5| 72.4 | 76.0 | 82.7 | 84.1 |
> > | -$\eta$=0.5 | 61.9 | 71.1 | 72.7 | 75.8 | 82.5 | 84.3 |
> >
> >
> > It is evident that the DPH-RL parameters must be adjusted to a specific order of magnitude; otherwise, weights that are too small fail to influence the gradients. We found that the weight for JS divergence should be around $10^{-1}$, while the optimal weight for FKL and RKL is $10^{-2}$. We hope future researchers can directly benefit from this finding. Within the $10^{-1}$ magnitude, variations in $\eta$ (e.g., increasing from 0.01 to 0.2 for DPH-JS) do not lead to performance collapse. DPH-JS demonstrates significant robustness, showing stable performance without major fluctuations across the range of 0.05 to 0.5.
> >
> > Response to Q2:
> > | $\eta$ | Reverse KL(Greedy) |
> > |:---|:-----------:|
> > | 0.005 | 59.7 |
> > | 0.01 | 60.0 |
> > | 0.02 | 58.7 |
> > | 0.05 | 57.6 |
> > | 0.2 | 53.8 |
> >
> > To ensure the RKL baseline was competitive, we detailed the performance of RKL under different $\eta$ values in Figure 2(b) and selected the optimal parameter $\eta=0.01$ as our reference. Our primary objective is to maximize Pass@k while surpassing the Pass@1 performance of GRPO (both with and without KL). Larger $\eta$ values caused a sharp decline in Pass@1, indicating that those parameters were excessive.
> >
> > **Q3: Can the method be applied to off-policy or batch RLHF settings, where distribution mismatch is more severe?**
> >
> > Yes, the DPH-RL framework is naturally adaptable to off-policy or batch settings. We base this on two key points:
> >
> > 1.  **Generalizable Training Objective**: As defined in Eq. (4), our general optimization goal is $max_{\pi_{\pi}}E_{q\sim D}[E_{a\sim\pi_{\pi}(\cdot|q)}[r(a|q)]-\eta D_{f}(\pi_{\pi}(\cdot|q)||\pi_{ref}(\cdot|q))]$. This objective is not restricted to on-policy rollouts; the data distribution $D$ can effectively represent a static offline dataset (batch RL) or a replay buffer (off-policy).
> >
> > 2.  **Transferability of the "Decoupling" Strateg**:
> > The core mechanism of DPH-RL (data partitioning) is highly transferable to offline settings. In this setup, $D_{exp}$ serves as the static dataset for policy optimization, while $D_{pef}$ utilizes high-quality samples to act as an "anchor" for FKL/JS regularization. This separation effectively mitigates distribution mismatch: while the policy learns from the offline data, it actively forces the policy to maintain coverage of the reference model's valid regions, thereby preventing the model from drifting into out-of-distribution (OOD) areas.
> >
> > The application of $f$-divergences in offline settings has already been established by methods such as $f$-DPO [2] and $f$-PO [3], and we have discussed these methods in the **Introduction** (p.1, lines 90-91) and **Related Work** (p.2, lines 140-143) sections of our paper
> >
> > [1] Chan A, Silva H, Lim S, et al. Greedification operators for policy optimization: Investigating forward and reverse kl divergences[J]. Journal of Machine Learning Research, 2022, 23(253): 1-79.
> >
> > [2] Wang C, Jiang Y, Yang C, et al. Beyond Reverse KL: Generalizing Direct Preference Optimization with Diverse Divergence Constraints[C]//The Twelfth International Conference on Learning Representations.
> >
> > [3] Han J, Jiang M, Song Y, et al. $ f $-PO: Generalizing Preference Optimization with $ f $-divergence Minimization[J].

---

> ### Author Response · Authors · 2025-11-27
> **Official Comment by Authors**
>
> We are very grateful for your thoughtful and detailed feedback on our manuscript. We have prepared a detailed response.
>
> As the discussion period is drawing to a close, please do not hesitate to let us know if there are any further questions or areas for clarification. We are ready and willing to discuss any remaining concerns. We thank you once more for your invaluable contribution to our work.

---

### Official Review · Reviewer_jdRw · 2025-11-11

**Soundness:** 3
**Presentation:** 3
**Contribution:** 2
**Rating:** 4
**Confidence:** 3

**Summary:**

This paper investigates the diversity collapse problem in RLVR for LLM post-training. The authors identify that the commonly used reverse-KL divergence is a key contributor to this problem, as its mode-seeking nature forces policies to converge on single high-probability solutions, leading to degraded Pass@k performance and catastrophic forgetting. To address this, the paper proposes DPH-RL, which leverages mass-covering f-divergences as a "rehearsal mechanism" to maintain broad solution coverage. Experiments on SQL generation and mathematical reasoning tasks with Llama and Qwen models demonstrate that DPH-RL improves both Pass@1 and Pass@k metrics while mitigating catastrophic forgetting on out-of-domain tasks.

**Strengths:**

- The paper targets an important limitation of current RLVR methods: the degradation of Pass@K performance despite improvements in Pass@1.

- The paper provides a comprehensive analysis of diversity collapse from the perspective of the KL divergence term, with both theoretical and empirical evidence.

- The method is orthogonal to existing approaches, allowing for potential combinations and broader applicability.

**Weaknesses:**

- The improvements are modest, and no single DPH variant consistently outperforms the baselines, which limits practical usage, i.e., different variants achieve best results on different datasets, making it unclear which configuration to use in practice.

- Although the paper's motivation is to address Pass@K degradation in RLVR, the proposed method does not convincingly alleviate this problem. Besides, the evaluation is limited to Pass@64, and larger K values (e.g., Pass@256, Pass@1024 done in [1]) are not reported. Such evaluations would better demonstrate whether the method truly preserves diversity at scale.

- The data partitioning setting lacks justification and ablation studies. The correctness threshold differs across models (6/8 for Llama, 7/8 for Qwen) without a clear explanation or ablation studies. It is unclear how robust the method is to different threshold choices.

[1] Does Reinforcement Learning Really Incentivize Reasoning Capacity in LLMs Beyond the Base Model?

**Questions:**

N/A

---

> ### Author Response · Authors · 2025-11-21
> **Response to Reviewer jdRw (Part 1)**
>
> Dear Reviewer jdRw,
>
> We sincerely appreciate your constructive feedback. In this revision, we have clarified the evaluation baseline standards and conducted the requested experiments on larger $k$ values (up to 256) and ablation studies on the correctness threshold. Ultimately, we identified DPH-JS (Generator) - (Threshold='8 of 8') as the consistently superior configuration for practical usage.
>
> Note: In our paper, 'Greedy' refers to Pass@1 at a temperature of 0.
>
> ## 1. Clarification: Definition of "Baselines" and Evaluation Focus
> *Reviewer Concern: The improvements are modest, and no single DPH variant consistently outperforms the baselines.*
>
> First, we must clarify that the **Baselines** in our paper specifically refer to **GRPO, DAPO, and Reverse KL (RKL)**, not the pre-RL Base Model. Our core objective is to **surpass all these baselines simultaneously on Pass@1 (accuracy) and Pass@k (diversity)**.
>
> **Regarding the Base Model:**
> While the Base Model typically exhibits high Pass@k (diversity), its Pass@1 (accuracy) is often too low for practical deployment (e.g., only 42.4% on SQL tasks). In this context, the primary goal of RL is to significantly boost Pass@1 (e.g., our method improves it by nearly 20%) [1-3]. As noted in the paper you cited [4], Post-RL models typically outperform base models by large margins in greedy decoding.
>
> **Our Contribution:**
> As shown in the table below, **our DPH-JS method substantially outperforms GRPO, DAPO, and RKL across every dimension.**
>
> | Model | Bird Greedy | Bird Pass@8 | Bird Pass@16 | Spider Greedy | Spider Pass@8 | Spider Pass@16 | OOD Dataset (Avg) |
> | :--- | :---: | :---: | :---: | :---: | :---: | :---: | :---: |
> | GRPO | 58.5 | 66.2 | 67.7 | 73.0 | 79.5 | 80.6 | 52.37 |
> | DAPO | 60.0 | 67.2 | 69.0 | 71.1 | 75.3 | 76.7 | 52.63 |
> | RKL | 60.0 | 69.8 | 71.8 | 71.0 | 79.0 | 80.6 | 48.45 |
> | DPH-F | 60.4 | 70.1 | 71.6 | 77.4 | 84.5 | 85.7 | **60.98** |
> | **DPH-JS** | **62.4** | **70.1** | **71.9** | **78.6** | **85.2** | **86.8** | 60.23|
>
>
> ## 2. Response on Larger K Values
> *Reviewer Concern: "Evaluation is limited to Pass@64... larger K values (e.g., Pass@256, Pass@1024) are not reported."*
>
> We have conducted the requested experiments extending the evaluation to Pass@256, where performance convergence was observed (see Figure 2(a), p.8, lines 400-406 in the revision). It is worth noting that while some studies evaluate Pass@1024 on the AIME benchmark (limited to only 30 problems), our evaluation is performed on the BIRD dataset (comprising 1,534 problems). Consequently, our results carry significantly stronger statistical power, and the observed convergence at Pass@256 is sufficient to demonstrate the method's effectiveness.
>
> | Model | pass@32 | pass@64 | pass@96 | pass@128 | pass@160 | pass@192 | pass@224 | pass@256 |
> | :---: | :---: | :---: | :---: | :---: | :---: | :---: | :---: | :---: |
> | Llama-3.1-8B-Instruct | 82.0 | 84.6 | 85.6 | 87.2 | 87.7 | 88.5 | 88.8 | 89.0 |
> | DPH-JS | 75.5 | 78.1 | 79.4 | 79.9 | 80.7 | 81.5 | 81.7 | 82.2 |
> | DPH-F | 75.2 | 77.8 | 79.6 | 79.6 | 79.9 | 80.9 | 81.5 | 81.5 |
> | RKL | 75.0 | 76.0 | 77.5 | 77.8 | 78.0 | 78.1 | 78.1 | 78.4 |
> | DAPO | 72.3 | 73.1 | 74.2 | 74.9 | 74.9 | 75.2 | 75.2 | 75.2 |
> | GRPO | 72.1 | 72.6 | 74.2 | 74.7 | 74.9 | 75.2 | 75.7 | 75.7 |
>
> The Pass@$k$ scores for all methods reach convergence by $k=256$. Notably, standard GRPO and DAPO saturate early, plateauing at approximately 75\%. Although DAPO outperforms GRPO at small $k$ values, its mechanism fails to scale effectively to larger $k$. The DPH family of methods significantly outperforms the GRPO baseline. Regarding Reverse KL, while it yields higher performance than GRPO, it remains significantly inferior to the DPH series. Overall, DPH-JS emerges as the superior method across both Pass@1 and Pass@$k$ metrics.

---

> ### Author Response · Authors · 2025-11-21
> **Response to Reviewer jdRw (Part 2)**
>
> ## 3. Response to Concern on Pass@K Degradation
>
> *Reviewer Concern: "Although the paper's motivation is to address Pass@K degradation in RLVR, the proposed method does not convincingly alleviate this problem."*
>
> We acknowledge that post-RL Pass@k degradation is a prevalent challenge in the field [4-6], representing the inherent trade-off between optimizing for pass@1 and maintaining the broad distribution of the pre-trained model. However, our extended experiments on **Larger K values** (up to $K=256$) provide compelling evidence that DPH significantly mitigates this issue compared to state-of-the-art baselines.
>
> As illustrated in the table above:
> 1.  **Baseline Collapse:** The baselines (GRPO, DAPO, and Reverse KL) exhibit a rapid saturation or "collapse" in performance as $K$ increases. Specifically, GRPO and DAPO plateau early at approximately ~75%, failing to unlock further potential even with more samples.
> 2.  **Effective Mitigation by DPH:** In stark contrast, the DPH series effectively inhibits this decline. DPH-JS maintains a robust upward trajectory, reaching **82.2%** at Pass@256, which is a substantial improvement over the ~75% ceiling of the baselines and much closer to the Base Model's theoretical upper bound.
>
> Therefore, while RL inevitably narrows the distribution to focus on correct answers, DPH proves to be significantly more effective than existing methods at preserving the diversity and reasoning potential of the model (Pass@k) while simultaneously achieving superior Pass@1.
>
> ## 4. Ablation Studies on the Correctness Threshold
> *Reviewer Concern: The data partitioning setting lacks justification and ablation studies.*
>
> We have updated the paper in Section 6.2. **'8 of 8'** indicates a threshold where samples correctly solved in 8 out of 8 attempts are assigned to $D_{\text{pef}}$ in DPH-JS. 'Full Data' implies using the entire dataset for both $D_{\text{pef}}$ and $D_{\text{exp}}$.
>
> | Model | Size of $\mathcal{D}_{\text{exp}}$ | Size of $\mathcal{D}_{\text{pef}}$ | Bird Greedy | Bird Pass@8 | Bird Pass@16 | Spider Greedy | Spider Pass@8 | Spider Pass@16 |
> | :---: | :---: | :---: | :---: | :---: | :---: | :---: | :---: | :---: |
> | GRPO | 8958 | 0 | 58.5 | 66.2 | 67.7 | 73.0 | 79.5 | 80.6 |
> | DAPO | 8958 | 0 | 60.0 | 67.2 | 69.0 | 71.1 | 75.3 | 76.7 |
> | 8 of 8 | 8266 | 692 | 61.4 | 71.6 | 73.5 | 75.4 | 86.3 | 87.2 |
> | 7 of 8 | 7429 | 1529 | 61.3 | 71.2 | 73.6 | 75.1 | 85.8 | 87.2 |
> | 6 of 8 | 6710 | 2248 | **62.8** | 70.5 | 72.4 | **76.0** | 82.7 | 84.1 |
> | 5 of 8 | 5783 | 3175 | 60.2 | 70.6 | 74.0 | 74.3 | 86.2 | 88.9 |
> | Full Data | 8958 | 8958 | 57.4 | **73.1** | **76.6** | 71.8 | **88.4** | **90.3** |
>
> Robustness Analysis: Our ablation studies show negligible performance differences between the '7 of 8' and '8 of 8' settings, indicating the method is robust to threshold selection. While the '6 of 8' setting can slightly boost greedy performance by focusing exploration on harder samples, further reduction (e.g., '5 of 8') degrades results due to the loss of training data. Conversely, the 'Full Data' setting yields poor greedy performance (worse than the baseline) despite high Pass@k. This failure stems from two issues: (1) Statistical Bias: Including rare successes (e.g., 1/8 correct) preserves noise rather than high-quality diversity; and (2) Trivial Optimization: The model minimizes loss by simply mimicking the single correct answer without genuine learning.
>
> Although we excluded simpler samples in the '6 of 8' setting to focus exploration on challenging problems and maximize greedy performance, for general purposes, we recommend '8 of 8' as the default strategy. Despite allocating only ~8% of the training data to $\mathcal{D}_{\text{pef}}$ (compensated for by retaining all sampled trajectories), it consistently outperforms GRPO and DAPO. Crucially, samples that are correctly solved 8 out of 8 times typically yield zero gradients in standard GRPO due to a lack of variance within the group. Our method effectively capitalizes on these high-quality, "mastered" samples that are otherwise wasted by GRPO, offering a statistically unbiased and simple solution.

---

> > ### Author Response · Authors · 2025-11-21
> > **Response to Reviewer jdRw (Part 3)**
> >
> > ## 5. Response to "Inconsistent Variants" and Practical Usage
> >
> > *Reviewer Concern: "No single DPH variant consistently outperforms the baselines... making it unclear which configuration to use."*
> >
> > In Tables 1–3 of the paper, all DPH methods ($\text{DPH-F}$, $\text{DPH-JS}$, and the $\alpha$-divergence variants) significantly outperform the baseline methods $\text{GRPO}$, $\text{DAPO}$, and $\text{Reverse KL}$. In the general setting, we have identified **$\text{DPH-JS}$ (Generator) - (Threshold = '8 of 8')** as the robust default configuration for practical applications.
> >
> > It achieves significantly higher Greedy performance compared to $\text{DPH-F}$. In our supplementary Larger $K$ experiments, the $\text{DPH-JS}$ curve also predominantly lies above that of $\text{DPH-F}$. Functionally, the Generator formulation is computationally efficient as it requires sampling only from the reference policy ($\pi_{\text{ref}}$). In contrast, the standard Divergence definition decomposes JS divergence into two components, necessitating sampling from both the current policy ($\pi$) and $\pi_{\text{ref}}$, which inevitably increases computational overhead by requiring the reference model during the forward pass.
> >
> > Furthermore, the '8 of 8' threshold represents the simplest strategy while remaining highly robust. Notably, these fully correct samples typically result in zero gradients within the $\text{GRPO}$ group advantage calculation (as there is no variance within the group); our method effectively capitalizes on this otherwise underutilized high-quality data.
> >
> > We have revised the paper to explicitly recommend DPH-JS (Generator) - (Threshold='8 of 8') as the optimal configuration for future researchers (p10, line 536-537).
> >
> > [1] Shao, Zhihong, et al. "DeepSeek-Math: Pushing the Limits of Mathematical Reasoning in Open Language Models." arXiv preprint arXiv:2402.03300.
> >
> > [2] Cobbe, Karl, et al. "Training verifiers to solve math word problems." arXiv preprint arXiv:2110.14168.
> >
> > [3] Yue Y, Chen Z, Lu R, et al. Does reinforcement learning really incentivize reasoning capacity in llms beyond the base model?[J]. arXiv preprint arXiv:2504.13837, 2025.
> >
> > [4] Song Y, Kempe J, Munos R. Outcome-based exploration for llm reasoning[J]. arXiv preprint arXiv:2509.06941, 2025.
> >
> > [5] Zhu X, Xia M, Wei Z, et al. The surprising effectiveness of negative reinforcement in LLM reasoning[J]. arXiv preprint arXiv:2506.01347, 2025.
> >
> > [6] Gao L, Schulman J, Hilton J. Scaling laws for reward model overoptimization[C]//International Conference on Machine Learning. PMLR, 2023: 10835-10866.

---

> ### Author Response · Authors · 2025-11-27
> **Official Comment by Authors**
>
> We are very grateful for your thoughtful and detailed feedback on our manuscript. We have prepared a detailed response.
>
> As the discussion period is drawing to a close, please do not hesitate to let us know if there are any further questions or areas for clarification. We are ready and willing to discuss any remaining concerns. We thank you once more for your invaluable contribution to our work.

---

### Author Response · Authors · 2025-11-25
**Summary of Rebuttal by Authors**

Dear Reviewers and Area Chairs,

We sincerely thank the reviewers for their insightful and constructive feedback. These comments have helped us clarify ambiguities and significantly improve the quality of our paper. We have revised the manuscript based on most suggestions, with all changes highlighted in **red**. Additionally, we have provided detailed responses to all questions raised by the reviewers. The following is a summary of our responses:


## Summary of Revisions

* **For Reviewer jdRw:** We have **clarified the definitions of our baselines** and conducted **additional experiments**, including Pass@$K$ evaluations with a larger $K$ and ablation studies on data partitioning settings. These results demonstrate that our method, DPH-JS, **significantly outperforms** baselines such as GRPO, DAPO, and Reverse KL.

* **For Reviewer 3S5Q:** We have **articulated the distinctions** between our method and the work of Chan et al. [1]. Furthermore, we reproduced the method from Chan et al. and demonstrated that DPH-JS **substantially outperforms** their "hard forward KL" approach. Regarding questions on hyperparameters, we have provided point-by-point responses referencing the tuning experiments already presented in the paper.

* **For Reviewer 8NDo:** We have addressed the motivation and analysis of the Forward KL by **drawing upon the diversity experiments in the Appendix and the pre- and post-RL comparisons in Section 6**. We also **clarified the distinction** between DPH and a standard replay buffer. Furthermore, we emphasized that the partitioning of the $D_{pef}$ dataset aims to **maintain diversity rather than accuracy**, a claim we have now supported with corresponding experimental evidence.

* **For Reviewer NVcc:** We included ablation studies on data partitioning settings to demonstrate that our method **remains robust even under data imbalance**. Based on these results, we recommend the "8 of 8" setting as the most generalizable configuration. Additionally, we provided a detailed explanation regarding the **stability of the JS divergence** and the **validity of our theoretical assumptions**.

* **For Reviewer HqbP:** We similarly utilized the data partitioning ablation studies to **alleviate concerns regarding extreme data scenarios**. We also confirmed that experiments were completed using $\alpha$-divergence and provided a theoretical analysis demonstrating that the DPH method can be **effectively extended to off-policy settings**.

[1] Chan A, Silva H, Lim S, et al. Greedification operators for policy optimization: Investigating forward and reverse kl divergences[J]. Journal of Machine Learning Research, 2022, 23(253): 1-79.

We look forward to receiving the reviewers' further feedback.

Best

---

### Author Response · Authors · 2025-11-28
**Global Response**

Dear Reviewers,

With the discussion period concluding in a few days, we are following up to ensure that our rebuttal (submitted last week) has fully addressed your concerns.

We believe our detailed responses and new results have resolved the major issues raised in the initial reviews. Could you please confirm if there are any remaining questions or barriers to raising your score?

With only a few days left, we hope for a brief engagement to clarify any lingering doubts.

Best regards, Authors

---

> ### Comment · Area_Chair_ps75 · 2025-11-28
> **AC response.**
>
> I have sent a reminder to all reviewers through a message visible only to the reviewer group. I will do my best to ensure the quality and integrity of the review process. If any issues arise, please feel free to contact me.
>
> best,
> AC.

---

> > ### Author Response · Authors · 2025-11-28
> > **Thanks**
> >
> > I really appreciate your help.

---

### Author Response · Authors · 2025-12-01
**Summary of Reviewers' Concerns and Our Rebuttal**

Dear Area Chair,

Thank you for the time and effort you have devoted to evaluating our submission, the reviews, and our rebuttal.

To help reduce your workload, we provide a brief summary of the key points raised by the reviewers, along with an overview of how our rebuttal addresses their concerns and the resulting improvements to the manuscript.

### Summary of Strengths
* **Clear Motivation & Theoretical Grounding (Reviewer jdRw, 3S5Q, NVcc, HqbP):** We provide a comprehensive analysis of diversity collapse caused by the KL divergence term, supported by both theoretical derivation and empirical evidence. Our work addresses a core issue regarding KL divergence usage in this field.
* **Simplicity & Efficiency (Reviewer jdRw, 3S5Q, NVcc, HqbP):** The proposed method is simple to implement and highly efficient, as it eliminates the need to load a reference model during training. Our method is orthogonal to existing approaches and can be easily integrated.
* **Strong Empirical Support (Reviewer 3S5Q, HqbP):** The claims are vigorously supported by experimental results across two different models (LLaMA and Qwen) and two Tasks (SQL and Math).

---

### Response to Common Concerns

**1. Pass@k performance relative to the Pre-RL model (Reviewers jdRW, 8NDo):**

The degradation of Pass@k under large sampling counts is a known issue [1]. Our objective is to mitigate this decline in Pass@k after RL training by utilizing a KL-divergence perspective, and to emphasize that DPH consistently outperforms all post-RL baselines (GRPO, DAPO, Reverse KL) on both Pass@1 and Pass@k.

**2. Robustness of Dataset Partitioning and Thresholds (Reviewer jdRw, 8NDo, NVcc, HqbP):**

We added ablation studies in Section 6.2 demonstrating strong robustness to threshold selection with negligible performance differences; we also suggest oversampling for imbalanced data and recommend the fixed '8 of 8' setting to eliminate tuning requirements.

**3. Applicability to Offline RLHF (Reviewers 3S5Q, HqbP):**

We explain that DPH-RL is naturally adaptable to offline/batch RLHF by applying its data decoupling strategy to offline datasets and using $D_{\text{pef}}$ as an anchor for $f$-divergence regularization to prevent policy drift.

---

### Response to Specific Reviewer Concerns

**For Reviewer jdRw:** We added experiments with larger $k$ sizes in Figure 2(a) (p.8, lines 400-406) and provided a detailed comparison of variants, concluding that DPH-JS is the optimal choice.

**For Reviewer 3S5Q:** We distinguish our work from [2] based on the optimization objective, algorithmic derivation, and implementation paradigm. Crucially, the definitions in Table 1 of [2] link Forward KL to **rejection sampling** and Reverse KL to **Reinforce** (which aligns with GRPO in our baseline). This association **fundamentally distinguishes** our methodology from that presented in [2]. We have performed additional experiments demonstrating that DPH outperforms Hard Forward KL in [2]. Additionally, we also pointed to existing coefficient settings in Table 1 and added Reverse KL tuning comparisons with 5 coefficients in Figure 2.

**For Reviewer 8NDo:** We reinforced our motivation via Appendix A and Section 4.3, and clarified in Section 6.1 that while simple samples preserve accuracy, partitioning is essential to prevent the diversity collapse observed in standard GRPO.

**For Reviewer NVcc:** We have updated the figures in the main text to improve clarity and resolution, and have responded to and clarified all questions regarding specific formulas.

**For Reviewer HqbP:** We added a comparison with Alpha-Divergence in Table 1 and clarified that maintaining high Pass@k is crucial for long-term Pass@1 optimization, as early diversity collapse hinders the model's ability to distinguish samples.
We hope this summary assists in your decision-making process.

Best regards,

The Authors

[1] Does Reinforcement Learning Really Incentivize Reasoning Capacity in LLMs Beyond the Base Model?

[2] Chan A, Silva H, Lim S, et al. Greedification operators for policy optimization: Investigating forward and reverse kl divergences[J]. Journal of Machine Learning Research, 2022, 23(253): 1-79.

---

### Meta-Review · Area_Chair_89LG · 2026-01-05

**Summary:**

This paper studies why reinforcement learning with verifiable rewards often has lower diversity in solutions (pass@k degrades). It proposes to use an alternative divergence functions, such as forward KL or JS divergence to better preserve diverse solutions during training. Results on math and SQL tasks show consistent improvement over standard baselines for pass@1 and pass@k with additional benefits of mitigating catastrophic forgetting up to some extend. Reviewers had different opinion if the contribution is significant with the respect to prior art and if improvements are statistically significant. Given positive reviews from 2 reviewers accepting the paper, I recommend acceptance.

**Reviewer Concerns:**

Authors provided extra ablation studies on larger top K, threshold and more. They also clarified the differences with prior forward-KL works.  However, the practical significance of results might be not clear after the rebuttal.

**Reviewer Scores:**

Reviewer NVcc (8 -> 8): Likely the same.
Reviewer jdRw (4 -> 5): Likely an increase given more ablations.
Reviewer 3S5Q (4 -> 5): Likely an increase after prior work comparison and ablations.
Reviewer 8NDo (4 -> 4): same as the significance of improvements has not being discussed.
Reviewer HqbP (8 -> 8): the same

---

### Decision · Program_Chairs · 2026-01-26

Accept (Poster)